# IMPACT: Industrial Machine Perception via Acoustic Cognitive Transformer

## Abstract

Industrial acoustic signals encode machine state, yet prevailing data-driven approaches are task-specific supervised pipelines that generalize poorly beyond their design conditions. Progress is further limited by the scarcity of large-scale datasets and pretrained models tailored to active shop floor audio. To address this, we introduce **DINOS** (**D**iverse **IN**dustrial **O**peration **S**ounds), a dataset of 74,149 recordings totaling over 1,093 hours collected from active manufacturing lines across diverse processes and operating regimes. We also provide **IMPACT** (**I**ndustrial **M**achine **P**erception via **A**coustic **C**ognitive **T**ransformer), a reference model pretrained on DINOS to standardize evaluation. Our benchmark is structured in four machine-specific steps: (1) *baseline discrimination*, (2) *moderate operational complexity*, (3) *scalability to unseen equipment*, and (4) *domain shift and sensor modality adaptation*. Across tasks, models pretrained or fine-tuned on DINOS outperform general-purpose audio models, demonstrating the value of domain-specific pretraining for industrial acoustic perception.

## 1 Introduction

Acoustic signals generated by industrial equipment carry critical operational insights for anomaly detection, predictive maintenance, and process optimization, which are essential components for improving system reliability and operational efficiency in manufacturing (Lee et al., 2024). While data-driven approaches have advanced this field, most of them rely on task-specific supervised learning, requiring domain-specific labeling efforts that limit their scalability across diverse industrial environments. Furthermore, the scarcity of publicly available large-scale datasets and pretrained models tailored to real-world industrial sounds hinders community-driven research, reproducibility, and further progress. Foundation models, which have achieved notable success in natural language processing (Devlin et al., 2019; Brown et al., 2020) and computer vision (Dosovitskiy et al., 2020; Radford et al., 2021; Kirillov et al., 2023), offer a promising path forward. Through large-scale self-supervised learning, these models can learn versatile representations that transfer effectively to downstream tasks with minimal labeled data (Zhang et al., 2024).

In the audio domain, similar approaches have emerged (Hershey et al., 2017; Huang et al., 2022; Elizalde et al., 2023), yet they primarily focus on general audio like music, YouTube videos, and conversational speech. However, industrial sounds differ from general-purpose audio. They often include stable tonal harmonics tied to machine kinematics, specific broadband noise profiles generated by physical processes, and unique temporal structures such as operational periodicity or diagnostically significant transients arising from faults (Randall, 2021; Bies et al., 2023; Antoni, 2007). As a result, models trained on general-purpose audio fail to capture these structured acoustic signatures, leading to reduced diagnostic sensitivity in industrial monitoring. To address these challenges, this paper presents two resources to shift this landscape (Figure 1).

First, we release **DINOS** (**D**iverse **IN**dustrial **O**peration **S**ounds), a large-scale open-access dataset covering diverse acoustic scenarios in manufacturing contexts. It comprises 74,149 recordings totaling over 1,093 hours across diverse manufacturing processes. In contrast to earlier datasets (e.g., MIMII (Purohit et al., 2019), ToyADMOS (Koizumi et al., 2019)), which focus on limited machine types or controlled faults in miniature parts, DINOS reflects operational diversity and real-world conditions. We employ two sensor types—microphone and stethoscope sensors—to capture both distinctive and blended acoustic characteristics of machines.

Figure 1: **System Overview.** The pipeline begins with dataset curation for DINOS, collecting industrial sounds from live shop floors using stethoscope and microphone sensors. Unlabeled data are used for pretraining and fine-tuning of models. Benchmarking evaluates performance on labeled data.

Second, we introduce **IMPACT** (**I**ndustrial **M**achine **P**erception via **A**coustic **C**ognitive **T**ransformer), a reference pretrained model based on a transformer architecture using self-supervised learning on DINOS. It provides a baseline for industrial sound foundation models. Additionally, we present benchmarking systems and results evaluating established sound foundation models. The benchmark follows a four-step progression: (1) *baseline performance on simple tasks*; (2) *sensitivity to periodic patterns under moderate operational complexity*; (3) *scalability via limited-data training on new equipment*; and (4) *adaptability on unseen domains* Models pretrained or fine-tuned on DINOS outperform general-purpose counterparts, demonstrating the need for domain-specific datasets and models in industrial acoustics. Through DINOS and IMPACT, we aim to facilitate community-driven research in industrial sound analysis.

Our contributions are summarized as follows:

- Release of DINOS, the first large-scale dataset of real shop floor machine sounds.
- A standardized benchmark framework spanning classification and anomaly detection tasks.
- IMPACT, a reference model pretrained on DINOS for standardized comparison.
- Baseline results across multiple models to support reproducibility and future research.

The remainder of the paper is structured as follows: Section 2 reviews related work, Section 3 details our dataset collection, Section 4 model training methodology, Section 5 presents benchmarking details and results, and Section 6 concludes with discussions on implications and future directions.

## 2 RELATED WORK

### 2.1 FROM GENERAL-PURPOSE AUDIO TO INDUSTRIAL ACOUSTICS

**From classical to deep learning** Acoustic machine condition monitoring has long been a key topic in industrial machine monitoring. Early approaches relied on hand-crafted features such as Mel-Frequency Cepstral Coefficients (MFCCs) with Gaussian Mixture Models (GMMs) or one-class Support Vector Machines (SVMs) (Chu et al., 2009; Sivasankaran & Prabhu, 2013; Heittola et al., 2013). With deep learning, autoencoders became a dominant framework; (Marchi et al., 2017) showed that recurrent autoencoders can model normal machine patterns and detect anomalies via reconstruction error. When labeled fault types are available, supervised learning has been applied to tasks such as tool-wear detection (Yun et al., 2023) and diagnostics for Additive Manufacturing (AM) (Lee et al., 2024). These methods often achieve high accuracy within the training domain but require substantial annotation and are typically trained per machine, limiting scalability and generalization to new operating conditions.

**General-purpose audio corpora** In general acoustic analysis, large public datasets have driven progress in general acoustic analysis. AudioSet (Gemmeke et al., 2017) provides over two million labeled 10-second clips from YouTube across 527 categories. ESC-50 and UrbanSound8K (Piczak,

2015; Salamon et al., 2014) focus on environmental sounds, with UrbanSound8K containing 8,732 short urban clips ($\leq 4$ s) in 10 classes. However, these corpora emphasize everyday audio (speech, music, ambient noise) and lack the structured signatures characteristic of industrial acoustics.

**General-purpose pretrained audio models**  As in vision and language, audio analysis has shifted from hand-crafted features to pretrained and self-supervised learning (SSL) models using large datasets. CNN-based VGGish (Hershey et al., 2017) and PANNs (Kong et al., 2020) learn general-purpose embeddings from large corpora. Transformer-based models advanced this trend: AST (Gong et al., 2021) applies patch-based self-attention to spectrograms, outperforming CNNs on ESC-50 (95.6% accuracy) and achieving competitive results on AudioSet (mAP 0.485). AudioMAE (Huang et al., 2022) introduced masked autoencoding for transfer across tasks. Speech models such as wav2vec 2.0 (Baevski et al., 2020) and HuBERT (Hsu et al., 2021) demonstrate strong representation learning with limited supervision. Yet these models are trained on everyday audio, which mismatches the physics and temporal structure of industrial sounds.

**Domain-specific datasets for machine anomaly detection**  Open industrial datasets emerged with MIMII (Purohit et al., 2019) and ToyADMOS (Koizumi et al., 2019). MIMII records four machine types (valves, pumps, fans, slide rails) under normal and anomalous conditions; ToyADMOS uses miniature machines with synthetic faults to provide labeled anomalies. MIMII-DG (Dohi et al., 2022) introduces domain shifts to probe robustness. Despite their impact on Anomalous Sound Detection (ASD), research, these datasets cover limited machine classes, often rely on artificial anomalies, and remain insufficient in diversity and scale to support pretraining for real shop floor acoustics.

**Domain-specific foundation models beyond industry**  Motivated by the limitations of general models, several works developed specialized foundation models for non-industrial audio domains. For instance, OPERA (Zhang et al., 2024) pretrained a transformer model on 400 hours of respiratory audio from coughs and breathing events, outperforming general-purpose models on 16 out of 19 medical acoustic tasks and illustrating the benefits of domain-specific pretraining. Similar domain-focused efforts have appeared in ecology and environmental monitoring (Chasmai et al., 2024; Piczak, 2015). However, to our knowledge, no prior foundation model has been trained specifically on real industrial sounds collected from active shop floors.

## 2.2 LIMITATIONS AND CONTRIBUTIONS

In summary, the field faces three pressing limitations. First, existing industrial sound datasets are narrow in scope, limited in scale, and insufficient for training foundation models. Second, no publicly available foundation model exists for industrial machine sound, leaving researchers to rely on task-specific solutions or general audio embeddings ill-suited to the domain. Third, current methods often fail to generalize across machines or environments due to strong domain dependence. To address these challenges, we present three contributions: (1) DINOS, a large-scale dataset from diverse industrial machines at live production sites; (2) a standardized benchmark framework with results for industrial machine monitoring; and (3) IMPACT, a reference pretrained model on DINOS designed to facilitate reproducible comparison. Together, these contributions aim to establish a scalable foundation for industrial machine listening and enable broader adoption of domain-specific research.

## 3 DINOS: DATASET CONSTRUCTION

### 3.1 DATA ACQUISITION

We collect industrial machine sound data using two sensor types: a microphone (Fifine K053) and a customized stethoscope sensor (Fifine K053 + MDF Instruments). The microphone captures both machine sounds and surrounding noise near a machine, which in confined industrial spaces often causes reverberation and crosstalk. In contrast, the stethoscope sensor effectively attenuates high-frequency ambient noise and better isolates localized machine sounds (Kim et al., 2025). By leveraging their complementary properties, we capture both global and localized acoustic characteristics of industrial machines. All recordings are made at 48,000 Hz, mono, 16-bit resolution to ensure high-fidelity signal acquisition.

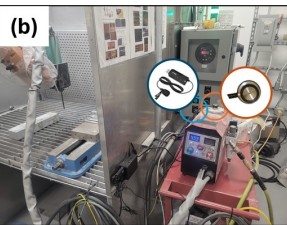 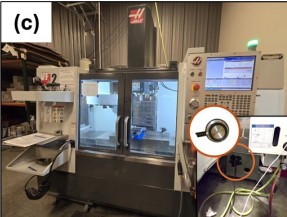 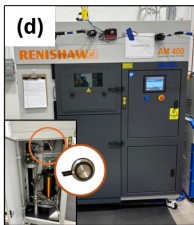

Figure 2: **Target Machines and Sensor Placements.** microphone: blue circle; stethoscope: red circle. (a) Yornew VMC-300 CNC: Microphone beside spindle; stethoscope on beneath table. (b) BaltiCold Spray LTD, CSM 108.2: Stethoscope and microphone on feeder. (c) Haas VF-2 CNC: Stethoscope on lower right base. (d) Renishaw AM400: Stethoscope inside powder handling panel.

## 3.2 DATA SOURCE AND DISTRIBUTION

Figure 2 illustrates representative sensor mounting. Specifically, mounting the stethoscope sensor on rigid structures improves coupling and avoids the damping and spectral distortion that can arise when attaching to flexible sheet-metal panels, yielding more consistent machine-specific acoustic signatures. The resulting dataset (Table 1) spans diverse manufacturing processes, materials, and operating conditions.

For cutting, we record sound from two CNC machines: Haas VF-2 and Yornew VMC-300. VF-2 includes three operational states (inactivity, machining, warm-up). Yornew machines aluminum (Al-6060) under varying feed rates and spindle speeds to induce chatter—a self-excited vibration that degrades surface finish and tool life by exciting the system's natural frequencies. We also record metal-processing audio from an APEC SK2540 CNC machine.

For metal AM processes, we record idle and operational states from Renishaw Laser Powder Fed Fusion (LPBF) and FormAlloy Directed Energy Deposition (DED) systems, capturing acoustic signatures of fan activation, axis motion, and laser operation. The Renishaw recordings come from two physically distinct units of the same model, denoted RenishawR and RenishawL; they were recorded and curated as separate datasets. A subset of RenishawR is included in the pretraining corpus, whereas RenishawL is reserved exclusively for the benchmarking tasks to ensure strict separation between pretraining and evaluation. For Cold Spray equipment, we record both normal operation and multiple anomalies, including gas-flow loss, powder clogging, and powder depletion. Finally, a microphone on a multi-machine shop floor records ambient industrial acoustics, including concurrent machine operations, fan rotation, and intermittent high-pressure air bursts.

Table 1: **DINOS Dataset.** DINOS comprises 74,149 samples totaling over 1,093 hours, providing a comprehensive reference for developing and benchmarking diagnostic and monitoring systems across diverse industrial acoustic environments. (Durations indicate per-sample length)

| Category | Sensor type | Samples | Duration | Distribution (Samples) | Category | Sensor type | Samples | Duration | Distribution (Samples) |
|---|---|---|---|---|---|---|---|---|---|
| CNC (SK2540) | Stethoscope | 21,570 | 59 s | Pretraining (1,851) | AM-LPBF (RenishawL) | Stethoscope | 524 | 1 s | Benchmarking (524) |
| AM-LPBF (RenishawR) | Stethoscope | 21,600 | 59 s | Pretraining (1,851) | CNC (VMC-300) | Stethoscope | 461 | 1 s | Benchmarking (461) |
| AM-DED (FormAlloy) | Stethoscope | 21,600 | 59 s | Pretraining (1,851) | CNC (VMC-300) | Microphone | 461 | 1 s | Benchmarking (461) |
| Shop floor | Microphone | 1,851 | 59 s | Pretraining (1,851) | AM-ColdSpray | Stethoscope | 2,455 | 1 s | Benchmarking (2,455) |
| CNC (VF-2) | Stethoscope | 1,118 | 1 s | Benchmarking (1,118) | AM-ColdSpray | Microphone | 2,509 | 1 s | Benchmarking (2,509) |

# 4 IMPACT: REFERENCE INDUSTRIAL SOUND FOUNDATION MODEL

## 4.1 PRETRAINING DATASETS

To empirically validate the effectiveness of DINOS and support its contribution, we train three IMPACT variants under an identical pretraining pipeline and input preprocessing. Prior to pretraining for all three versions, samples are processed using Root Mean Square (RMS) normalization ($x_{\mathrm{rms}} = \sqrt{\frac{1}{n} \sum_{i=1}^{n} x_i^2}$) and Z-score normalization ($z = \frac{x - \mu}{\sigma}$) methods, and then segmented into 1-second clips while maintaining the sampling rate, resolution, and bit depth.

**IMPACT-DCASE.** Trained exclusively on the DCASE2025 Challenge Task 2 datasets (Harada et al., 2021; Dohi et al., 2022), this variant serves as a baseline reflecting performance achievable with publicly available industrial-sound corpora.

**IMPACT-DINOS.** Trained solely on our DINOS dataset, this is the primary model in this work. To avoid sampling bias, we selected an equal number of recordings from four DINOS categories (1,851 per category; 7,404 recordings in total), amounting to 121 hours of audio (Table 1). We included contact stethoscope recordings to emphasize localized, structure-borne machine signatures and shop floor microphone recordings to capture blended, dynamic industrial acoustics.

**IMPACT-Hybrid.** Trained on the union of the DCASE2025 data and DINOS, this variant tests whether combining heterogeneous sources improves generalization.

## 4.2 ARCHITECTURE

As shown in Figure 3, IMPACT adopts a student–teacher masked-autoencoding framework inspired by EAT (Chen et al., 2024). Each 1-s sound clip is converted to a log-Mel spectrogram $\mathbf{x} \in \mathbb{R}^{1 \times 128 \times 128}$.

**CNN encoder and patching.** A CNN encoder maps $\mathbf{x}$ to $\mathbf{F} \in \mathbb{R}^{C \times 64 \times 64}$ with $C{=}32$. We partition $\mathbf{F}$ into non-overlapping $16 \times 16$ patches to obtain a $4 \times 4$ grid (thus $N{=}16$ patches). Each patch is flattened and linearly projected to a $d$-dimensional token ($d{=}384$), then augmented with fixed 2D positional encodings. Let $\mathbf{Z} \in \mathbb{R}^{N \times d}$ denote the token sequence.

**Student branch.** The student operates on a masked version $\tilde{\mathbf{Z}}$ by replacing $40\%$ of tokens with a learned mask token. Let $\mathbf{h}_s^{\text{CLS}} \in \mathbb{R}^d$ be the student's [CLS] token from the last Transformer layer.

**Teacher branch.** The teacher processes the unmasked spectrogram through the same encoder and Transformer. Let $\mathbf{H}_t^{(\ell)} \in \mathbb{R}^{N \times d}$ be the token matrix at Transformer layer $\ell \in \{1, \dots, L\}$. We first average across layers to obtain $\bar{\mathbf{H}}_t = \frac{1}{L} \sum_{\ell=1}^{L} \mathbf{H}_t^{(\ell)} \in \mathbb{R}^{N \times d}$, then mean-pool over tokens to obtain the teacher's global vector $\mathbf{g}_t = \frac{1}{N} \sum_{i=1}^{N} \bar{\mathbf{H}}_{t,i} \in \mathbb{R}^d$. The utterance-level alignment loss is $\mathcal{L}_u = \|\mathbf{h}_s^{\text{CLS}} - \mathbf{g}_t\|_2^2$.

**CNN decoder.** For frame-level reconstruction, student tokens are projected as $\mathbf{z}_i' = \phi(\mathbf{W}_p \mathbf{z}_i + \mathbf{b}_p) \in \mathbb{R}^{256}$ and $\tilde{\mathbf{z}}_i = \mathbf{W}_d \mathbf{z}_i' + \mathbf{b}_d \in \mathbb{R}^{128}$, where $\phi$ is ReLU. Arranging $\{\tilde{\mathbf{z}}_i\}_{i=1}^{16}$ on the $4 \times 4$ grid yields a feature map $\mathbf{U} \in \mathbb{R}^{128 \times 4 \times 4}$, which is passed through the CNN decoder layers. Let $\hat{\mathbf{x}}_s \in \mathbb{R}^{1 \times 128 \times 128}$ be the student reconstruction; the teacher branch symmetrically produces $\hat{\mathbf{x}}_t$. The frame-level loss is defined as a distillation loss between reconstructions, $\mathcal{L}_f = \|\hat{\mathbf{x}}_s - \hat{\mathbf{x}}_t\|_2^2$.

**Objective and EMA update.** The total loss is $\mathcal{L}_{\text{total}} = \mathcal{L}_f + \lambda \mathcal{L}_u$ with $\lambda{=}0.20$. Teacher parameters are updated via an exponential moving average of the student after each training step with decay $\tau{=}0.9999$; no gradients flow into the teacher.

## 5 BENCHMARKING

### 5.1 BENCHMARK DATASETS

Table 2 summarizes four machine-specific datasets spanning 27 downstream tasks. No benchmark sample is included in the pretraining data. Each dataset targets a distinct evaluation objective: (i) **RenishawL**—binary activity detection (on/off); (ii) **VF2**—classification of overall machining activity (on/off) and the warm-up cycle; (iii) **Yornew**—fine-grained 12-way classification over machining states that vary by cutting depth (CD), material removal rate (MRR), revolutions per minute (RPM), and chatter; and (iv) **ColdSpray**—eight-way classification derived from four operational states (normal, depleted powder, powder clogging, no gas supply) measured with two sensor modalities (microphone and stethoscope), enabling both within- and cross-modality evaluation. We also define anomalous sound detection (ASD) tasks: for Yornew, chatter-present states are labeled abnormal and chatter-absent states normal; for ColdSpray, the three fault conditions are grouped as the abnormal class against normal.

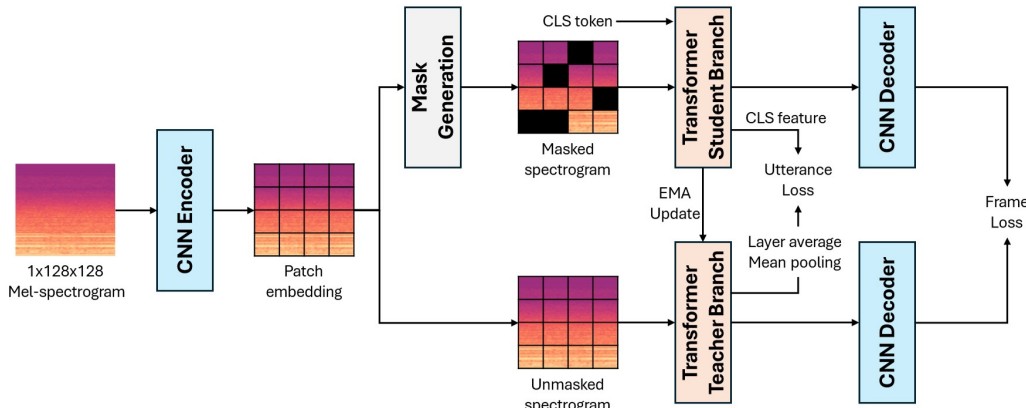

Figure 3: **Architecture of IMPACT.** Overall pipeline with student-teacher branches. The student is trained using a reconstruction and alignment objective, while the teacher is updated via EMA. The detailed hyperparameters are presented in Appendix A.1.3.

For every task, we adopt a Monte Carlo cross-validation scheme: 15% of the samples are randomly selected for training and the remaining 85% for evaluation. To reduce variance and sampling bias, we repeat this procedure ten times with independent random splits (stratified by class). We maintain strict separation between pretraining data, benchmark training sets, and benchmark evaluation sets. Final metrics are reported as the mean and standard deviation across the ten runs.

Table 2: **Benchmarking Dataset.** The dataset comprises 27 downstream tasks covering binary and multi-class classification across four machine types. It is designed to evaluate robustness to operational variation, sensor modality, and domain shift.

| Machine | ID | Description / Parameter | Samples | Machine | ID | Description / Parameter | Samples |
|---|---|---|---|---|---|---|---|
| RenishawL | T1 | On | 386 | Yornew | T15 | CD: 3.0, MRR: 76.2, RPM: 12K, Chatter: N | 78 |
| RenishawL | T2 | Off | 138 | Yornew | T16 | CD: 3.0, MRR: 76.2, RPM: 12K, Chatter: Y | 76 |
| VF2 | T3 | On | 260 | Yornew | T17 | CD: 3.0, MRR: 76.2, RPM: 8K, Chatter: N | 76 |
| VF2 | T4 | Off | 504 | ColdSpray | T18 | Normal (stethoscope) | 1,214 |
| VF2 | T5 | Warm-up | 354 | ColdSpray | T19 | Depleted powder (stethoscope) | 472 |
| Yornew | T6 | CD: 0.3, MRR: 7.62, RPM: 8K, Chatter: N | 78 | ColdSpray | T20 | Powder clogging (stethoscope) | 385 |
| Yornew | T7 | CD: 0.3, MRR: 7.62, RPM: 12K, Chatter: N | 78 | ColdSpray | T21 | No gas supply (stethoscope) | 384 |
| Yornew | T8 | CD: 0.3, MRR: 7.62, RPM: 4K, Chatter: N | 78 | ColdSpray | T22 | Normal (microphone) | 1,268 |
| Yornew | T9 | CD: 0.5, MRR: 12.7, RPM: 12K, Chatter: N | 74 | ColdSpray | T23 | Depleted powder (microphone) | 472 |
| Yornew | T10 | CD: 0.5, MRR: 12.7, RPM: 8K, Chatter: N | 76 | ColdSpray | T24 | Powder clogging (microphone) | 385 |
| Yornew | T11 | CD: 0.5, MRR: 12.7, RPM: 4K, Chatter: N | 78 | ColdSpray | T25 | No gas supply (microphone) | 384 |
| Yornew | T12 | CD: 1.0, MRR: 25.4, RPM: 4K, Chatter: Y | 78 | Yornew | T26 | Normal/Abnormal | 618 / 304 |
| Yornew | T13 | CD: 1.0, MRR: 25.4, RPM: 8K, Chatter: N | 78 | ColdSpray | T27 | Normal/Abnormal | 2482 / 2482 |
| Yornew | T14 | CD: 1.0, MRR: 25.4, RPM: 12K, Chatter: Y | 74 | | | | |

## 5.2 BENCHMARK BASELINES & METRICS

Along with the IMPACT models, we benchmark four widely used pretrained audio models, one toolkit, one fine-tuned variant, and two domain-specific models. **OpenSMILE** (Eyben et al., 2010) is a feature-extraction toolkit for speech and audio; we use the ComParE configuration, a standardized set commonly employed in paralinguistics and affective computing tasks (Schuller et al., 2016). **CLAP** (Elizalde et al., 2023) is a multimodal model trained on paired audio–text data across diverse sources. **VGGish** (Hershey et al., 2017) is a CNN-based model derived from the VGG family and pretrained on YouTube audio via AudioSet, providing general-purpose embeddings widely used for audio classification. **AudioMAE** (Huang et al., 2022) is a transformer-based self-supervised model pretrained via masked autoencoding (on AudioSet/ESC-50/Speech Commands/VoxCeleb) and has shown strong transfer across downstream audio tasks; we additionally include a fine-tuned AudioMAE variant trained on a subset of DINOS to assess the utility of domain-specific data. For domain-specific comparison, we evaluate **OPERA** (Zhang et al., 2024), a family of specialized foundation models for respiratory acoustics, using its two transformer variants (OPERA-CT and OPERA-GT; contrastive vs. generative objectives). Both OPERA models are designed to process stethoscope recordings, which aligns with a portion of our data acquisition.

We adopt two evaluation protocols by task type. For classification (T1–T25), we use a standard linear-probing setup: a single fully connected layer trained on top of frozen features, reporting mAP. For anomaly detection (T26–T27), we use a reconstruction-based method with an autoencoder head and report the Area Under the Receiver Operating Characteristic curve (AUROC). To ensure consistent comparison, all inputs follow a unified preprocessing pipeline with padding/truncation to the required length; when a baseline mandates a specific front end or input length, we follow its original implementation, keeping all other training details aligned with the respective official procedures.

## 5.3 Benchmark Details and Results

We evaluate all models on 27 downstream tasks across four industrial machines (Table 2). Evaluation follows two protocols by task type: mAP for classification tasks (T1–T25) and AUROC for anomalous sound detection (ASD) tasks (T26–T27). Aggregate results are shown in Table 3, and per-task results in Table 4.

**RenishawL (T1–T2): On/Off Classification of LPBF System.** This binary classification task establishes a baseline for evaluating each model's ability to distinguish between distinct machine states. Most deep learning-based models achieved near-perfect performance (mAP $\approx$ 1.0000), whereas the conventional feature-based OpenSMILE produced poor results (mAP = 0.5000), demonstrating the advantage of learned representations.

**VF2 (T3–T5): CNC operation mode classification.** This multi-class setting evaluates active, idle, and warm-up states under moderate operational complexity. IMPACT-DCASE achieves the highest average mAP (0.9771), but the margin among IMPACT variants is small ($< 0.004$). Transformer-based models all exceed 0.93 mAP on average, while the CNN baseline VGGish lags behind (mAP = 0.8736). This suggests that transformer architectures are generally more effective than CNNs for industrial sound analysis. At the same time, models pretrained on general audio corpora such as CLAP and AudioMAE-PreT. also reach strong performance, indicating that periodic states with moderate complexity including on/off/warm-up cycles can be learned from broad-domain pretraining, while domain-specific pretraining such as DINOS provides additional, consistent improvements.

**Yornew (T6–T17, T26): Fine-Grained State Recognition and Anomaly Detection.** These tasks assess the model's capacity to discriminate among 12 distinct CNC machining configurations and to detect chatter as an anomaly. The benchmark evaluates scalability to new equipment, not included in DINOS, by examining the handling of overlapping frequencies, transient variations, and limited samples. For the fine-grained classification task (T6–T17), IMPACT-Hybrid achieved the highest performance (mAP = 0.8971), implying the synergistic benefits of combining the DINOS and DCASE datasets for complex discrimination challenges. The performance of OPERA is also notable; given that the Yornew data were also collected with a stethoscope, this model likely benefits from an inductive bias tailored to structure-borne vibrations. Additionally, the superiority of OPERA-GT over OPERA-CT in this low-data scenario suggests that generative approaches, which capture the intrinsic data structure, may be more data-efficient. For anomaly detection (T26), the chatter detection task provides a practical evaluation of diagnostic capability across operational conditions. Here, IMPACT-DINOS yielded the best results (AUROC = 0.8041), indicating that pretraining on a focused, domain-specific dataset may enhance sensitivity in anomaly detection scenarios by minimizing interference from broader data sources.

**ColdSpray (T18–T25, T27): Generalization to Unseen Machines.** These tasks evaluate generalization to an unseen industrial domain—one that involves physics and phenomena entirely absent from the DINOS dataset—through classification of normal versus multiple fault states across both stethoscope and microphone sensors, as well as anomaly detection by grouping faults as abnormal, to assess adaptability to shifts in machine physics and sensor modalities. IMPACT-Hybrid achieves the second highest mAP (0.9277) and the highest AUROC (0.8041), showing robustness to domain and modality shifts. A particularly insightful finding comes from the OPERA models, which show strong performance on both sensor types. Their success, despite being trained on medical respiratory data, is likely because they are highly adept at capturing acoustic patterns related to fluid dynamics. The fault conditions in the ColdSpray process—such as powder clogging or gas supply loss—are fundamentally fluid flow anomalies, analogous to the respiratory events OPERA was trained to recognize. This unexpected success underscores the importance of curating pretraining datasets that cover not just

diverse machines, but also a wide range of underlying physical acoustic sources. Further evidence for the importance of data is the significant mAP gain of AudioMAE-FineT (from 0.7370 to 0.8370) after exposure to DINOS, affirming the dataset's role in adapting models to complex industrial sounds.

Table 3: **Overall Model Performance per Machine.** mAP is reported for classification tasks and AUROC for anomaly detection tasks. Bold indicates best performance.

| Machine | OpenSMILE | CLAP | VGGish | OPERA-CT | OPERA-GT | AudioMAE-PreT. | AudioMAE-FineT. | IMPACT-DCASE | IMPACT-DINOS | IMPACT-Hybrid |
|---|---|---|---|---|---|---|---|---|---|---|
| RenishawL (mAP) | 0.5000 ± 0.0000 | **1.0000 ± 0.0000** | 0.9891 ± 0.0108 | **1.0000 ± 0.0000** | **1.0000 ± 0.0000** | 0.9959 ± 0.0024 | 0.9866 ± 0.0194 | **1.0000 ± 0.0000** | **1.0000 ± 0.0000** | **1.0000 ± 0.0000** |
| VF2 (mAP) | 0.3556 ± 0.0349 | 0.9364 ± 0.0145 | 0.8736 ± 0.0208 | 0.9596 ± 0.0162 | 0.9594 ± 0.0126 | 0.9418 ± 0.0166 | 0.9655 ± 0.0196 | **0.9771 ± 0.0078** | 0.9746 ± 0.0108 | 0.9737 ± 0.0104 |
| Yornew (mAP) | 0.0832 ± 0.0003 | 0.8043 ± 0.0368 | 0.5461 ± 0.0256 | 0.8039 ± 0.0189 | 0.8516 ± 0.0275 | 0.5743 ± 0.0167 | 0.6890 ± 0.0338 | 0.8316 ± 0.0408 | 0.8841 ± 0.0231 | **0.8971 ± 0.0250** |
| ColdSpray (mAP) | 0.1271 ± 0.0045 | 0.8137 ± 0.0060 | 0.7771 ± 0.0079 | **0.9415 ± 0.0063** | 0.9297 ± 0.0062 | 0.7370 ± 0.0141 | 0.8370 ± 0.0100 | 0.8344 ± 0.0049 | 0.9125 ± 0.0070 | 0.9277 ± 0.0072 |
| Yornew (AUROC) | 0.5645 ± 0.0065 | 0.8778 ± 0.0292 | 0.7403 ± 0.0153 | 0.8828 ± 0.0140 | 0.8523 ± 0.0220 | 0.6224 ± 0.0148 | 0.7000 ± 0.0363 | 0.7777 ± 0.0238 | **0.9121 ± 0.0206** | 0.9077 ± 0.0181 |
| ColdSpray (AUROC) | 0.4635 ± 0.0119 | 0.7163 ± 0.0105 | 0.6824 ± 0.0083 | 0.7724 ± 0.0074 | 0.7220 ± 0.0112 | 0.4371 ± 0.0129 | 0.6316 ± 0.0186 | 0.7176 ± 0.0123 | 0.7852 ± 0.0064 | **0.8041 ± 0.0144** |

Table 4: **Per-Class Model Performance for Downstream Tasks.** mAP scores for classification (T1-T25) and AUROC for anomaly detection (T26-T27). Bold indicates best performance.

| Task | OpenSMILE | CLAP | VGGish | OPERA-CT | OPERA-GT | AudioMAE-PreT. | AudioMAE-FineT. | IMPACT-DCASE | IMPACT-DINOS | IMPACT-Hybrid |
|---|---|---|---|---|---|---|---|---|---|---|
| T1 | 0.7357 ± 0.0000 | **1.0000 ± 0.0000** | 0.9857 ± 0.0150 | **1.0000 ± 0.0000** | **1.0000 ± 0.0000** | 0.9987 ± 0.0009 | 0.9882 ± 0.0173 | **1.0000 ± 0.0000** | **1.0000 ± 0.0000** | **1.0000 ± 0.0000** |
| T2 | 0.2643 ± 0.0000 | **1.0000 ± 0.0000** | 0.9926 ± 0.0181 | **1.0000 ± 0.0000** | **1.0000 ± 0.0000** | 0.9931 ± 0.0036 | 0.9850 ± 0.0197 | **1.0000 ± 0.0000** | **1.0000 ± 0.0000** | **1.0000 ± 0.0000** |
| T3 | 0.2904 ± 0.0861 | 0.9720 ± 0.0074 | 0.9003 ± 0.0281 | **0.9856 ± 0.0040** | 0.9824 ± 0.0050 | 0.9366 ± 0.0202 | 0.9714 ± 0.0358 | 0.9828 ± 0.0045 | 0.9810 ± 0.0136 | 0.9802 ± 0.0108 |
| T4 | 0.4596 ± 0.0134 | 0.9211 ± 0.0337 | 0.8809 ± 0.0283 | 0.9477 ± 0.0219 | 0.9469 ± 0.0239 | 0.9163 ± 0.0259 | 0.9567 ± 0.0173 | **0.9686 ± 0.0139** | 0.9662 ± 0.0167 | 0.9662 ± 0.0160 |
| T5 | 0.3168 ± 0.0000 | 0.9161 ± 0.0118 | 0.8395 ± 0.0249 | 0.9455 ± 0.0242 | 0.9490 ± 0.0143 | 0.9725 ± 0.0085 | 0.9683 ± 0.0157 | **0.9797 ± 0.0067** | 0.9767 ± 0.0092 | 0.9746 ± 0.0079 |
| T6 | 0.0839 ± 0.0021 | 0.9093 ± 0.0956 | 0.7451 ± 0.0872 | 0.8947 ± 0.0632 | 0.9613 ± 0.0309 | 0.5663 ± 0.0761 | 0.8744 ± 0.0751 | 0.9796 ± 0.0326 | 0.9850 ± 0.0132 | **0.9860 ± 0.0165** |
| T7 | 0.0835 ± 0.0038 | 0.6842 ± 0.0839 | 0.2148 ± 0.0546 | 0.7349 ± 0.0772 | 0.7505 ± 0.0621 | 0.3421 ± 0.1334 | 0.5098 ± 0.1502 | 0.6573 ± 0.0937 | 0.7675 ± 0.0831 | **0.7863 ± 0.0683** |
| T8 | 0.0847 ± 0.0001 | 0.5057 ± 0.0932 | 0.2868 ± 0.0703 | 0.6324 ± 0.0611 | 0.6554 ± 0.0526 | 0.1983 ± 0.0559 | 0.3880 ± 0.0746 | 0.5373 ± 0.0545 | 0.6705 ± 0.0719 | **0.7007 ± 0.0741** |
| T9 | 0.0802 ± 0.0008 | 0.8383 ± 0.0808 | 0.6204 ± 0.0464 | 0.7486 ± 0.0707 | 0.8358 ± 0.0609 | 0.7242 ± 0.0990 | 0.4217 ± 0.1017 | 0.8687 ± 0.0713 | 0.9385 ± 0.0522 | **0.9410 ± 0.0318** |
| T10 | 0.0823 ± 0.0000 | 0.8515 ± 0.0884 | 0.6762 ± 0.1153 | 0.9389 ± 0.0132 | 0.9452 ± 0.0348 | 0.6566 ± 0.1059 | 0.5898 ± 0.1319 | 0.9185 ± 0.0557 | 0.9547 ± 0.0372 | **0.9644 ± 0.0421** |
| T11 | 0.0847 ± 0.0000 | 0.8524 ± 0.1008 | 0.4504 ± 0.1164 | 0.8358 ± 0.1397 | 0.8751 ± 0.1293 | 0.5614 ± 0.0890 | 0.6929 ± 0.1539 | 0.8475 ± 0.1214 | 0.9483 ± 0.1453 | **0.9553 ± 0.1279** |
| T12 | 0.0847 ± 0.0000 | 0.8706 ± 0.1282 | 0.5404 ± 0.1074 | 0.9198 ± 0.0367 | 0.9509 ± 0.0392 | 0.6029 ± 0.0500 | 0.8896 ± 0.1115 | 0.9203 ± 0.0819 | **0.9898 ± 0.0200** | 0.9886 ± 0.0217 |
| T13 | 0.0847 ± 0.0000 | 0.9574 ± 0.0193 | 0.7908 ± 0.0377 | 0.9354 ± 0.0525 | 0.9557 ± 0.0282 | 0.7754 ± 0.1239 | 0.9167 ± 0.0740 | 0.9785 ± 0.0138 | 0.9819 ± 0.0105 | **0.9924 ± 0.0056** |
| T14 | 0.0802 ± 0.0006 | 0.9654 ± 0.0210 | 0.6397 ± 0.0942 | 0.8970 ± 0.0549 | 0.9450 ± 0.0445 | 0.5045 ± 0.1076 | 0.6069 ± 0.1165 | 0.8788 ± 0.1238 | 0.9872 ± 0.0128 | **0.9898 ± 0.0100** |
| T15 | 0.0847 ± 0.0000 | **0.7856 ± 0.0655** | 0.5248 ± 0.0482 | 0.5963 ± 0.0836 | 0.6489 ± 0.0308 | 0.6452 ± 0.0426 | 0.6773 ± 0.0742 | 0.6987 ± 0.0908 | 0.7044 ± 0.0771 | |
| T16 | 0.0823 ± 0.0000 | 0.5714 ± 0.0954 | 0.5592 ± 0.0744 | 0.6315 ± 0.0825 | 0.7168 ± 0.1006 | 0.5199 ± 0.0807 | **0.8164 ± 0.0286** | 0.7527 ± 0.0956 | 0.7544 ± 0.0682 | 0.7820 ± 0.0688 |
| T17 | 0.0823 ± 0.0000 | 0.8600 ± 0.0451 | 0.5046 ± 0.0667 | 0.8815 ± 0.0653 | **0.9785 ± 0.0206** | 0.7918 ± 0.0848 | 0.9160 ± 0.0747 | 0.9628 ± 0.0343 | 0.9322 ± 0.0840 | 0.9741 ± 0.0351 |
| T18 | 0.2445 ± 0.0000 | 0.9426 ± 0.0092 | 0.8499 ± 0.0209 | 0.9836 ± 0.0061 | 0.9782 ± 0.0048 | 0.8640 ± 0.0103 | 0.9755 ± 0.0049 | 0.9502 ± 0.0095 | 0.9915 ± 0.0035 | **0.9932 ± 0.0033** |
| T19 | 0.0951 ± 0.0000 | 0.9586 ± 0.0121 | 0.9093 ± 0.0138 | 0.9828 ± 0.0080 | 0.9812 ± 0.0057 | 0.7424 ± 0.0272 | 0.9453 ± 0.0156 | 0.9200 ± 0.0115 | 0.9863 ± 0.0071 | **0.9925 ± 0.0046** |
| T20 | 0.0776 ± 0.0000 | 0.8959 ± 0.0148 | 0.7744 ± 0.0262 | 0.9642 ± 0.0087 | 0.9543 ± 0.0159 | 0.7376 ± 0.0617 | 0.9513 ± 0.0114 | 0.9243 ± 0.0113 | 0.9803 ± 0.0108 | **0.9851 ± 0.0114** |
| T21 | 0.0774 ± 0.0000 | 0.9308 ± 0.0241 | 0.8387 ± 0.0388 | 0.9886 ± 0.0055 | 0.9795 ± 0.0123 | 0.9854 ± 0.0057 | **1.0000 ± 0.0001** | 0.9951 ± 0.0034 | 0.9996 ± 0.0010 | 0.9991 ± 0.0013 |
| T22 | 0.2548 ± 0.0018 | 0.9171 ± 0.0059 | 0.8827 ± 0.0106 | 0.9734 ± 0.0043 | 0.9691 ± 0.0044 | 0.8446 ± 0.0123 | 0.9321 ± 0.0108 | 0.9399 ± 0.0079 | 0.9731 ± 0.0025 | **0.9766 ± 0.0025** |
| T23 | 0.1129 ± 0.0361 | 0.4718 ± 0.0302 | 0.4735 ± 0.0224 | **0.8401 ± 0.0199** | 0.7842 ± 0.0296 | 0.3452 ± 0.0203 | 0.4459 ± 0.0478 | 0.5578 ± 0.0340 | 0.6844 ± 0.0238 | 0.7443 ± 0.0341 |
| T24 | 0.0776 ± 0.0000 | 0.3930 ± 0.0207 | 0.4948 ± 0.0356 | **0.7995 ± 0.0334** | 0.7910 ± 0.0245 | 0.3770 ± 0.0135 | 0.4460 ± 0.0493 | 0.3882 ± 0.0228 | 0.6852 ± 0.0392 | 0.7311 ± 0.0263 |
| T25 | 0.0774 ± 0.0000 | **1.0000 ± 0.0000** | 0.9938 ± 0.0081 | **1.0000 ± 0.0000** | **1.0000 ± 0.0000** | 0.9999 ± 0.0001 | **1.0000 ± 0.0000** | **1.0000 ± 0.0000** | **1.0000 ± 0.0000** | **1.0000 ± 0.0000** |
| T26 | 0.5645 ± 0.0065 | 0.8778 ± 0.0292 | 0.7403 ± 0.0153 | 0.8828 ± 0.0140 | 0.8523 ± 0.0220 | 0.6224 ± 0.0148 | 0.7000 ± 0.0363 | 0.7777 ± 0.0238 | **0.9121 ± 0.0206** | 0.9077 ± 0.0181 |
| T27 | 0.4635 ± 0.0119 | 0.7163 ± 0.0105 | 0.6824 ± 0.0083 | 0.7724 ± 0.0074 | 0.7220 ± 0.0112 | 0.4371 ± 0.0129 | 0.6316 ± 0.0186 | 0.7176 ± 0.0123 | 0.7852 ± 0.0064 | **0.8041 ± 0.0144** |

## 6 CONCLUSION, LIMITATIONS, AND FUTURE WORK

The benchmark results demonstrate that general-purpose sound models struggle with industrial acoustic tasks, while domain-specific pretraining on DINOS yields substantial improvements. The effectiveness of DINOS in capturing these characteristics is evident from the performance gains in AudioMAE after fine-tuning on DINOS and the comparative results across IMPACT variants. These findings indicates the limitations of general models in handling unique features of industrial sound, including stable tonal harmonics linked to machine kinematics, broadband noise from physical processes, and temporal structures such as operational cycles or fault-induced transients. It demonstrates the necessity of datasets composed of real-world industrial sounds from live production environments to enable models to learn robust, transferable representations. The contributions of this work lie in releasing DINOS, a large-scale dataset derived from actual manufacturing sites; establishing a benchmark protocol that evaluates model performance using sounds from real equipment across diverse tasks; and introducing IMPACT variants as reference pretrained models to serve as a baseline for future research in industrial acoustic perception.

Despite these contributions, this study has several limitations. Although IMPACT-DINOS and IMPACT-Hybrid demonstrate robust results, the OPERA models show notable performance in Cold-Spray tasks despite being trained on medical respiratory data. This suggests that simply varying machine types is insufficient. Instead, datasets should be further constructed with a deeper understanding of the mechanisms by which industrial sounds are generated. For instance, friction-induced sounds from CNC machining, impact-driven sounds from bending and stamping processes, pulse-like sounds from welding, electromagnetic hums from motors and power systems, fluid flow noise from pneumatic or spraying operations, and transient sounds from fracture, rupture, or deformation all represent distinct acoustic signatures. Expanding DINOS with such physically grounded categories will allow models to generalize across machines not only by type but also by the underlying physical phenomena that govern sound generation.

Another limitation concerns sensor diversity. DINOS currently relies on microphones and stethoscope sensors, but industrial environments often employ varied sensing modalities, including accelerometers, laser microphones, or current probes. Sound characteristics are highly sensitive to sensor type and placement, and broader sensor coverage would enhance the robustness and transferability of learned representations. Additionally, data bias remains a challenge. The dataset captures a subset of production environments with relatively controlled noise conditions, while real-world manufacturing involves unpredictable background noise, spatial reverberations, and shifting acoustic conditions. Future datasets should incorporate multi-sensor recordings collected across diverse environments, complemented by domain randomization and augmentation techniques to mitigate such biases.

At the same time, we recognize that the current benchmarking method, while valuable as an early-stage investigation, is limited in both the number of tasks and the scale of labeled samples. To build a richer and more comprehensive benchmarking suite, future work will focus on systematically expanding task diversity and data coverage. A key direction is the integration of automatic labeling systems that leverage machine operating information (e.g., controller logs, sensor metadata, and process parameters) to generate labels with minimal manual intervention. Such a framework would allow continuous and scalable dataset updates, enabling DINOS to grow into a progressively richer resource for industrial acoustic perception research.

From a modeling perspective, although IMPACT is more lightweight than other baselines (18M parameters compared to VGGish at 62M and AudioMAE at 86M), it remains challenging to deploy on severely resource-constrained embedded devices. Model compression, quantization, and architecture distillation will be essential to enable real-time inference at the edge. Moreover, the current approach processes audio in fixed 1-second segments, which may fail to capture events that occur either over very short durations or across longer cycles. Developing architectures that can flexibly handle variable-length inputs would improve applicability to diverse monitoring tasks.

Finally, data sharing in industrial domains remains highly restricted due to security and intellectual property concerns, limiting access to diverse, large-scale datasets. To address this bottleneck, future work should pursue hybrid data strategies that combine real-world industrial recordings with physics-informed synthetic audio generated from digital twin systems. Such an approach would expand dataset diversity while preserving confidentiality. We also encourage community-driven contributions to expand DINOS with additional machine types, sensing modalities, and operational scenarios, fostering collective progress in industrial acoustic perception.

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

# A    APPENDIX

## A.1    TECHNICAL APPENDICES AND SUPPLEMENTARY MATERIAL

### A.1.1    MACHINE DESCRIPTION

**AM-LPBF: AM400 (Renishaw).**    The AM400 is a laser powder bed fusion (LPBF) metal Additive Manufacturing (AM) system developed by Renishaw. LPBF is a metal 3D printing process that uses a laser to selectively melt metal powder in layers under an inert gas atmosphere. It supports materials such as Inconel 718 (RenishawL) and 316 stainless steel (RenishawR). The build chamber is maintained under inert argon gas to prevent oxidation. The official build volume is $250 \times 250 \times 300$ mm.

**AM-DED: L2 Series (FormAlloy).**    The L2 Series by FormAlloy is a directed energy deposition (DED) system designed for high-deposition-rate metal additive manufacturing. DED is a metal 3D printing method that melts and deposits material simultaneously using a laser and metal powder or wire. It processes materials such as A709 structural steel and 316H stainless steel using a coaxial laser and powder nozzle system. In this system, three powder feeders are used simultaneously to perform layered deposition. The laser power can reach up to 8 kW. It supports multi-material deposition, enabling the fabrication of complex, compositionally graded parts.

**CNC: VMC300 (Yornew).**    The Yornew VMC-300 is a compact 5-axis CNC vertical milling center designed for prototyping and educational use. In this dataset, the machine is operated as a 3-axis CNC by detaching the A and C axes and is used to machine aluminum (Al-6060) material. A 1/4-inch diameter two-flute end mill (YG-1; 01047) is applied to the milling experiments. The machine provides a working volume of $300 \times 150 \times 100$ mm, with a maximum spindle speed of 24,000 rpm. The maximum feed rate is 2,000 mm/min, and the spindle motor is rated at 750 W.

**CNC: VF-2 (Haas).**    The Haas VF-2 is a high-performance 3-axis vertical machining center widely used in industrial and academic environments. For this study, a 1/2-inch two-flute end mill tool is utilized to machine ABS plastic. The VF-2 offers a working envelope of $762 \times 406 \times 508$ mm, with a maximum spindle speed of 8,100 rpm and feed rates up to 16.5 m/min. The machine is equipped with an automatic tool changer (ATC) that supports up to 20 tools.

**CNC: SK2540 (APEC CNC).**    Metal processing is performed on the real shop floor using the SK2540 model. This machine supports a working area of $4000 \times 2500 \times 1000$ mm. It provides a rapid traverse rate of 60 m/min (XY) and 40 m/min (Z), and a 5 m/s² acceleration on all axes. The spindle operates at up to 24,000 rpm with a maximum spindle power of 75 kW, making it suitable for high-speed, high-precision aerospace aluminum machining tasks.

**Coldspray: CSM 108.2 (BaltiCold Spray LTD).**    The CSM 108.2 is a cold spray system used for solid-state deposition of metallic powders. In the described experiment, copper (Cu) powders with a particle size range of 10–45 $\mu$m and a mean diameter (d50) of 17 $\mu$m are used. The powder is sprayed at room temperature using nitrogen gas at a constant gauge pressure of 0.7 MPa without preheating. The deposition process enables bonding without melting, preserving the original microstructure of the material. This system is commonly used for research on coating performance and defect detection under controlled conditions.

### A.1.2    FREQUENCY ANALYSIS

Figure 4 presents the mean frequency spectra with standard deviation represented as error bars, obtained by applying FFT to every 2048-sample segment of the full acoustic recordings from each manufacturing process.

Figure 4(a) illustrates the analysis of the Renishaw additive manufacturing process, where segments corresponding to active deposition are labeled as "on," and all other segments as "off." A clear spectral peak emerges around 3000 Hz, indicating the dominant operating frequency during the build phase. Figure 4(b) shows the frequency spectra of the VF-2 CNC machine under three distinct states: cutting, idle, and warm-up. Warm-up data are collected during non-cutting operations in which axis

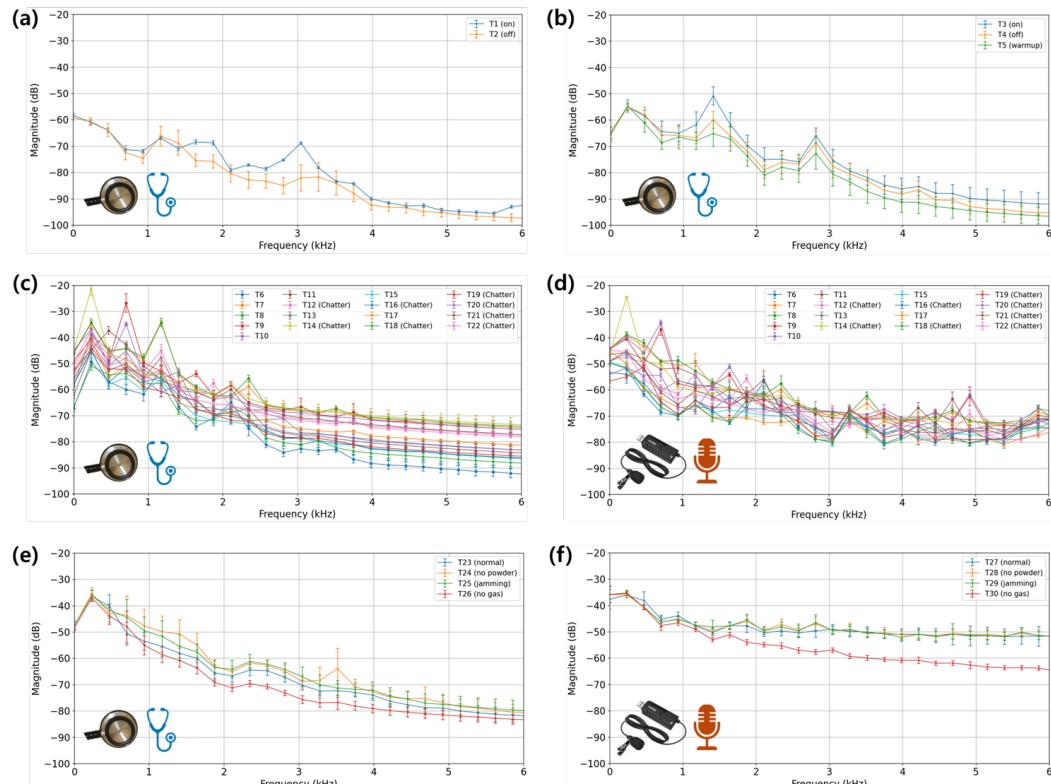

Figure 4: **Mean frequency spectra with standard deviation error bars for various machine operating states and sensor configurations.** (a) Frequency response of the Renishaw AM machine during operational and idle states, measured using a stethoscope acoustic sensor. (b) Spectra of the VF-2 CNC machine during operational, idle, and warm-up phases, also captured using a stethoscope. (c–d) Mean and variability of frequency spectra from the Yornew CNC machine at different cutting parameters and chatter conditions, recorded using (c) a stethoscope-type sensor and (d) a standard microphone. (e–f) Spectral characteristics of cold spray process anomalies—normal operation, powder absence, feed jamming, and gas cutoff—measured using (e) a stethoscope and (f) a microphone. All plots show the mean magnitude in decibels (dB) with standard deviation represented as error bars across frequency bins up to 6 kHz.

movements and spindle speed control are executed independently; the X-axis moved 304.8 mm, while the Y and Z axes moved 127 mm, alongside controlled acceleration of the spindle. Idle and non-cutting segments are categorized as "off," which included background noise such as pump and tool changer operation. Cutting data are recorded during the machining of ABS workpieces.

Figures 4(c) and 4(d) display acoustic data collected from the Yornew CNC machine cutting aluminum, using a stethoscope sensor and a conventional microphone, respectively. These plots reveal both global and local signal features: globally, the overall magnitude varies according to MRR, while locally, distinct spectral peaks increase with spindle speed and relate to the number of tool flutes. This indicates that both broadband and frequency-specific information encode key process parameters. These multi-sensor measurements are used to assess IMPACT's ability to classify dynamic changes in process conditions based on both sensor domains.

Figures 4(e) and 4(f) focus on the Cold Spray process, where acoustic signals from the stethoscope and microphone sensors are analyzed to evaluate IMPACT's capability to distinguish abnormal states independently. Compared to the normal case, the "no powder" condition produced high-frequency components near 1500 Hz and 3500 Hz, attributed to direct collisions between the vibration valve and powder feeder in the absence of powder. In the "jamming" state, insufficient damping from powder resulted in stronger structural vibration, leading to overall higher magnitude spectra. In contrast, the "no gas" condition—where powder ejection fails due to lack of compressed air—produced

significantly lower spectral magnitudes. The low-frequency amplification and high-frequency noise suppression characteristics of the stethoscope sensor are evident when comparing Figures 4(c) to 4(d) and Figures 4(e) to 4(f), respectively.

### A.1.3 IMPACT ARCHITECTURE AND HYPERPARAMETERS

The IMPACT model (Table 5) comprises three components: a CNN encoder, a Transformer encoder, and a CNN decoder.

**Patch embedding:** after converting each clip to a log-Mel spectrogram of size $1 \times 128 \times 128$ (FFT 2048, window 2048, hop 376, 128 Mel bands, top_dB 80), a Conv2d-based patch embed (kernel/stride 16×16) produces a $4 \times 4$ token grid ($N{=}16$) with embedding dimension $d{=}384$; a learnable [CLS] token is prepended and fixed 2D sinusoidal positional encodings are added to patch tokens.

**CNN encoder:** a single Conv2d+BN+*ReLU* (in: 1 ch, out: 32 ch, kernel 3×3, stride 2, padding 1) maps the input to $32 \times 64 \times 64$.

**Transformer encoder:** 8 layers with $d{=}384$, 16 heads, GELU activation.

**CNN decoder:** tokens are first projected $384{\rightarrow}256$ (encoder-to-decoder bridge) *and then* $256{\rightarrow}128$ per token before deconvolution; the deconvolution stack upsamples $128{\times}4{\times}4 \rightarrow 128{\times}8{\times}8 \rightarrow 64{\times}16{\times}16 \rightarrow 32{\times}32{\times}32 \rightarrow 16{\times}64{\times}64 \rightarrow 1{\times}128{\times}128$ using ConvTranspose2d+BN+ReLU (kernel $4{\times}4$, stride 2, pad 1) and a final ConvTranspose2d to 1 channel.

**Training details (summary):** mask ratio 0.4; utterance loss weight $\lambda{=}0.2$; teacher updated by per-step EMA with decay $\tau{=}0.9999$; optimizer AdamW (lr $5{\times}10^{-5}$, wd $10^{-5}$), batch size 128, 10 epochs; gradient clipping at 0.5; optional mixed precision. Frame-level loss is MSE between student and teacher reconstructions; utterance-level loss is MSE between the student's [CLS] and the teacher's global representation. All preprocessing and training were performed in PyTorch 2.8.0 on an Ubuntu 22.04.5 LTS system, equipped with an AMD Ryzen Threadripper Pro 7975WX, 128 GB RAM, and NVIDIA RTX A6000 Ada.

Table 5: **IMPACT architecture and hyperparameters.**

| CNN Encoder | Input Channels | Output Channels | Kernel | Stride | Padding |
|---|---|---|---|---|---|
| Conv2d + BN + ReLU | 1 | 32 | $3 \times 3$ | 2 | 1 |
| **Transformer Encoder** | #Layers | Dimension | #Heads | Activation | — |
| Transformer | 8 | 384 | 16 | GELU | — |
| **CNN Decoder** | Input Channels | Output Channels | Kernel | Stride | Padding |
| Linear (FC): $256{\rightarrow}128$ | — | — | — | — | — |
| ConvTranspose2d + BN + ReLU | 128 | 128 | $4 \times 4$ | 2 | 1 |
| ConvTranspose2d + BN + ReLU | 128 | 64 | $4 \times 4$ | 2 | 1 |
| ConvTranspose2d + BN + ReLU | 64 | 32 | $4 \times 4$ | 2 | 1 |
| ConvTranspose2d + BN + ReLU | 32 | 16 | $4 \times 4$ | 2 | 1 |
| ConvTranspose2d (Output) | 16 | 1 | $4 \times 4$ | 2 | 1 |
| **Linear probing head** | Input Dim | Output Dim | — | — | — |
| Linear (FC) + ReLU | 384 | 256 | — | — | — |
| Linear (FC) + ReLU | 256 | #classes | — | — | — |
| **Reconstruction head** | Input Dim | Output Dim | — | — | — |
| Linear (FC) + ReLU | 384 | 256 | — | — | — |
| Linear (FC) + ReLU | 256 | 128 | — | — | — |
| Linear (FC) + ReLU | 128 | 384 | — | — | — |

### A.1.4 ABLATION STUDY ON MASKING RATIO AND LOSS WEIGHT

To analyze the sensitivity of IMPACT to its key hyperparameters, we conducted an ablation study varying the masking ratio (MR $\in \{0.3, 0.4, 0.5, 0.6\}$) and the utterance-level loss weight $\lambda \in \{0.1, 0.2, 0.3\}$. Performance is reported in terms of mAP for classification tasks on ColdSpray,

RenishawL, Yornew, and VF2 datasets. As shown in Table 6, a masking ratio of 0.4 and 0.5 consistently yields the best performance, while 0.3 and 0.6 degrade results. Increasing $\lambda$ improves performance up to 0.2, after which the effect saturates or slightly declines. The radar plot in Figure 5 visualizes these trends across machines, showing that the configuration (MR = 0.4, $\lambda = 0.2$) provides the most balanced and robust results across all benchmark datasets. We therefore adopt this configuration in all main experiments.

Table 6: Ablation Study Results. Mean $\pm$ Std for classification (mAP) and anomaly detection (AUROC) across datasets.

| Dataset / Metric | MR=0.3 | | | MR=0.4 | | | MR=0.5 | | | MR=0.6 | | |
|---|---|---|---|---|---|---|---|---|---|---|---|---|
| | $\lambda = 0.1$ | $\lambda = 0.2$ | $\lambda = 0.3$ | $\lambda = 0.1$ | $\lambda = 0.2$ | $\lambda = 0.3$ | $\lambda = 0.1$ | $\lambda = 0.2$ | $\lambda = 0.3$ | $\lambda = 0.1$ | $\lambda = 0.2$ | $\lambda = 0.3$ |
| RenishawL (mAP) | $1.0000 \pm 0.0000$ | $1.0000 \pm 0.0000$ | $1.0000 \pm 0.0000$ | $1.0000 \pm 0.0000$ | $1.0000 \pm 0.0000$ | $1.0000 \pm 0.0000$ | $1.0000 \pm 0.0000$ | $1.0000 \pm 0.0000$ | $1.0000 \pm 0.0000$ | $1.0000 \pm 0.0000$ | $1.0000 \pm 0.0000$ | $1.0000 \pm 0.0000$ |
| VF2 (mAP) | $0.9756 \pm 0.0078$ | $0.9761 \pm 0.0080$ | $0.9775 \pm 0.0087$ | $0.9703 \pm 0.0124$ | $0.9746 \pm 0.0108$ | $0.9761 \pm 0.0110$ | $0.9383 \pm 0.0184$ | $0.9764 \pm 0.0095$ | $0.9746 \pm 0.0112$ | $0.9674 \pm 0.0142$ | $0.9708 \pm 0.0123$ | $0.9762 \pm 0.0101$ |
| Yornew (mAP) | $0.8817 \pm 0.0258$ | $0.8839 \pm 0.0239$ | $0.8853 \pm 0.0266$ | $0.8792 \pm 0.0249$ | $0.8841 \pm 0.0231$ | $0.8867 \pm 0.0233$ | $0.8453 \pm 0.0235$ | $0.8850 \pm 0.0220$ | $0.8888 \pm 0.0220$ | $0.8323 \pm 0.0290$ | $0.8823 \pm 0.0247$ | $0.8882 \pm 0.0230$ |
| ColdSpray (mAP) | $0.8941 \pm 0.0069$ | $0.8954 \pm 0.0071$ | $0.8975 \pm 0.0071$ | $0.9043 \pm 0.0085$ | $0.9125 \pm 0.0070$ | $0.9079 \pm 0.0070$ | $0.8841 \pm 0.0077$ | $0.9046 \pm 0.0078$ | $0.9075 \pm 0.0098$ | $0.8853 \pm 0.0092$ | $0.8713 \pm 0.0067$ | $0.8823 \pm 0.0101$ |
| Yornew (AUROC) | $0.9057 \pm 0.0174$ | $0.9073 \pm 0.0184$ | $0.9124 \pm 0.0202$ | $0.9060 \pm 0.0174$ | $0.9121 \pm 0.0206$ | $0.9102 \pm 0.0208$ | $0.9107 \pm 0.0193$ | $0.9174 \pm 0.0191$ | $0.9084 \pm 0.0186$ | $0.9027 \pm 0.0190$ | $0.9168 \pm 0.0180$ | $0.9165 \pm 0.0189$ |
| ColdSpray (AUROC) | $0.7671 \pm 0.0082$ | $0.7647 \pm 0.0081$ | $0.7632 \pm 0.0064$ | $0.7848 \pm 0.0098$ | $0.7852 \pm 0.0064$ | $0.7730 \pm 0.0097$ | $0.7862 \pm 0.0129$ | $0.7723 \pm 0.0088$ | $0.7733 \pm 0.0080$ | $0.7739 \pm 0.0102$ | $0.7615 \pm 0.0166$ | $0.7721 \pm 0.0117$ |

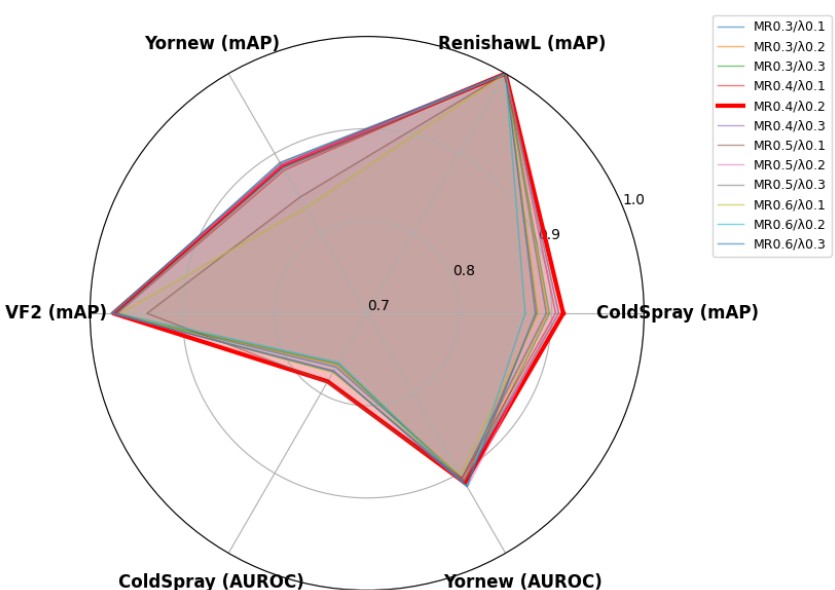

Figure 5: Radar plot of the ablation study results for IMPACT under different masking ratios (MR) and loss weightings ($\lambda$). Performance is reported across six benchmark metrics: RenishawL (mAP), VF2 (mAP), Yornew (mAP, AUROC), and ColdSpray (mAP, AUROC). Each curve corresponds to a specific (MR, $\lambda$) configuration, with the highlighted red curve (MR = 0.4, $\lambda = 0.2$) representing the best-performing setting. Results show that performance remains consistently high across settings, while intermediate masking and balanced weighting provide the most robust results across all machines and task types.

**Ethics Statement**    This work adheres to the ICLR Code of Ethics. All data used in this study were collected from industrial machinery in controlled laboratory and production environments without involvement of human subjects, thereby avoiding issues related to personal privacy or consent. The released dataset is anonymized and contains only machine acoustic signals, ensuring no sensitive or personally identifiable information. To mitigate risks of misuse, we document intended research purposes and clearly describe the scope of applications in Sections "Dataset" and "Benchmarking." We acknowledge potential dual-use concerns where diagnostic models may be deployed in safety-critical systems; to address this, we emphasize reproducibility, transparency, and proper evaluation protocols, and we encourage further community auditing. No conflicts of interest or external sponsorships influenced this work.

**Reproducibility Statement**    We aim to make this work easy to reproduce. Dataset sources, sensor configurations, and per-category counts are summarized in Table 1, while the downstream benchmark tasks and sample counts are detailed in Table 2. The IMPACT architecture is depicted in Figure 3, and its layer-wise specifications and training hyperparameters are enumerated in Table 5 (see Appendix, "IMPACT architecture and hyperparameters"). Our preprocessing pipeline (RMS and Z-score normalization) and pretraining corpora for each variant are described in Section "IMPACT: Reference Industrial Sound Foundation Model" under "Pretraining datasets." Benchmark protocols—including the Monte Carlo cross-validation procedure (15% train / 85% eval, 10 independent stratified splits) and evaluation metrics—are given in Section "Benchmarking," with aggregate results in Table 3 and per-task results in Table 4. We additionally report controlled sensitivity analyses in the ablation study (Table 6) and visualize cross-task trends in Figure 5. To facilitate verification and extension, we will release a repository containing the IMPACT implementation, training and evaluation scripts, and the exact benchmark splits, along with documentation describing data preprocessing and configuration files used to produce the reported numbers.

