# OpenReview forum: "IMPACT: Industrial Machine Perception via Acoustic Cognitive Transformer"
_ICLR.cc/2026/Conference — Submitted to ICLR 2026_

### Official Review · Reviewer_66g5 · 2025-10-16

**Soundness:** 3
**Presentation:** 2
**Contribution:** 2
**Rating:** 2
**Confidence:** 4

**Summary:**

This paper introduces a new dataset DINOS, consisting of 74149 acoustic samples collected from active manufacturing lines. The authors then proposed a pretraining method, IMPACT, which is conceptually similar to EAT. The authors evaluated its performance on 27 downstream tasks using DINOS.

**Strengths:**

1. The collection of DINOS is an earnest effort. DINOS consists of the signals collected from both a microphone and a stethoscope, and covers various types of equipment.
2. The authors evaluated the performance of various off-the-shelf pretrained models on DINOS.

**Weaknesses:**

1. The evaluation is critically insufficient and cannot show the superiority of IMPACT. The authors did not apply other pretraining methods (e.g., AudioMAE) on DINOS. They only evaluated the off-the-shelf pretrained models (e.g., a model pretrained using AudioMAE method on other acoustic datasets) on DINOS. Since IMPACT is a pretraining method, if the authors want to show the superiority of IMPACT, they need to **pretrain** IMPACT and other pretraining methods (e.g., AudioMAE) **on the same datasets**.
2. The proposed pretraining method is conceptually not sufficiently novel. Its similarity to EAT is also acknowledged by the authors.
3. Therefore, if the majority of the contributions lie in the introduction of DINOS, then this paper might be below the bar of ICLR. It might be more suitable to submit this paper to a venue specialized in industrial sensing or a venue offering dataset tracks.
4. The presentation could be improved overall.

**Questions:**

Please see weaknesses

---

> ### Author Response · Authors · 2025-11-23
>
> We sincerely appreciate your thoughtful review. While we value your recognition of the earnest effort in curating DINOS, we would like to offer a different perspective regarding evaluation sufficiency and venue suitability. Our intention is not merely to release a dataset, but to provide a foundational study that establishes methodology, benchmark standards, and a reference model for the field of industrial acoustic perception.
>
> ---
>
> **1. Foundational Contribution: Establishing the Standard for Industrial Acoustic Perception**
>
> Regarding the concern that the contribution may lie primarily in the introduction of DINOS, we agree that DINOS is the most visible component of our work. However, our main contribution is to address a fundamental gap: the absence of a coherent, standardized, and physically grounded foundation for industrial acoustics. Currently, industrial sound research largely relies on small or simplified datasets (e.g., MIMII) and heterogeneous evaluation protocols, making it difficult to assess how modern SSL models behave under realistic shop-floor conditions. Our work addresses this gap in three complementary ways:
>
> - **Dataset from Real-World Industrial Scenes (DINOS):** DINOS is, to our knowledge, the first large-scale industrial acoustic dataset captured directly from active industrial environments using both microphones and stethoscopes. It provides the scale (1,093 hours) and physical fidelity needed for realistic industrial AI evaluation.
> - **Reference Model (IMPACT):** Without a domain-tailored baseline, it is difficult to interpret performance numbers or measure progress. Therefore, we provide IMPACT as a reference foundation, enabling future work to answer “How good is my model relative to a strong, industrially optimized baseline?”. This parallels the role of widely adopted reference backbones in various domains, which serve as stable baselines for measuring progress.
> - **Structured hierarchical benchmark protocol:** We introduce a 4-step, 6-type hierarchical benchmark that evaluates models across increasing levels of difficulty and domain mismatch from simple on/off discrimination to cross-machine and cross-domain generalization. To our knowledge, this is the first attempt to organize industrial acoustic evaluation along such a progression. It provides a physically meaningful rationale for assessing generalization rather than relying on a single matched-condition anomaly task.
>
> Beyond these pillars, our experiments also provide physics-informed dataset design insights. We observe that the diversity of underlying physical sound-generation mechanisms (e.g., frictional contact, fluid flow, impacts) matters more for generalization than simply increasing machine types. For example, the unexpectedly strong performance of OPERA (trained on respiratory, fluid-dominated acoustics) on ColdSpray suggests that similarity in physical acoustic patterns is often more important than similarity in machine identity. These insights are presented as design guidelines for future industrial audio datasets. Thus, the contribution goes beyond a dataset release and proposes a standard for how industrial acoustic perception should be evaluated and how data should be constructed.

---

> ### Author Response · Authors · 2025-11-23
>
> **2. Evaluation Fairness: Comparison with AudioMAE Trained from Scratch**
>
> To directly address your concern regarding fairness, we pretrained AudioMAE from scratch on DINOS in addition to the existing AudioMAE-FineT baseline. We compare three AudioMAE variants and our IMPACT-DINOS:
>
> | Model | RenishawL (mAP) | VF2 (mAP) | Yornew (mAP) | ColdSpray (mAP) | Yornew (AUROC) | ColdSpray (AUROC) |
> | :--- | :--- | :--- | :--- | :--- | :--- | :--- |
> | AudioMAE-PreT. (AudioSet) | 0.9959±0.0024  | 0.9418±0.0166  | 0.5743±0.0167  | 0.7370±0.0141  | 0.6224±0.0148  | 0.4371±0.0129  |
> | AudioMAE-FineT. (DINOS)    | 0.9866±0.0194  | 0.9655±0.0196  | 0.6890±0.0338  | 0.8370±0.0100  | 0.7000±0.0363  | 0.6316±0.0186  |
> | AudioMAE-Scratch (DINOS)    | 1.0000±0.0000  | 0.9482±0.0186  | 0.7729±0.0412  | 0.7932±0.0136  | 0.7238±0.0187  | 0.5221±0.0061  |
> | IMPACT-DINOS                   | 1.0000±0.0000  | 0.9746±0.0108  | 0.8841±0.0231  | 0.9125±0.0070  | 0.9121±0.0206  | 0.7852±0.0064  |
>
> From these results, we draw two main observations:
>
> - DINOS is an effective pretraining source. Both AudioMAE-Scratch and AudioMAE-FineT outperform AudioMAE-PreT, confirming that DINOS provides industrial-specific acoustic characteristics absent in generic web audio.
> - Despite having only ~18M parameters (vs. ~86M for AudioMAE), IMPACT delivers higher performance even under equal pretraining conditions:
>     - ColdSpray (unseen domain): 0.9125 (IMPACT-DINOS) vs. 0.8370 (AudioMAE-FineT.)  (+7.6% mAP)
>     - Yornew (chatter anomaly): 0.9121 (IMPACT-DINOS) vs. 0.7238 (AudioMAE-Scratch) (+18.3% AUROC)
>
> This demonstrates that IMPACT's performance is not due solely to access to DINOS, but to its domain-optimized hybrid architecture (Physics-aware Hybrid design). We will add a condensed version of this table and discussion to the revised manuscript.
>
> ---
>
> **3. Technical Novelty: Physics-Aware Architectural Advancements**
>
> You correctly point out that IMPACT is conceptually similar to EAT. We agree that IMPACT builds on the same general student–teacher, masked prediction philosophy, and we state this explicitly. We, however, do not intend to introduce a completely new SSL paradigm, but to adapt and extend this paradigm to industrial acoustics via a hybrid architecture that addresses practical and physical aspects of this domain. Concretely, IMPACT differs from EAT in several ways:
>
> - **Dual-stage CNN Encoding vs. Single-stage Patching:**
>     - **Physics-Aware Feature Extraction:** EAT uses a CNN encoder mainly as a non-overlapping patch embedder (kernel size equals stride), so each token is computed from a disjoint spectro-temporal region. IMPACT adds a CNN feature extractor with overlapping receptive fields before patch embedding. This design helps preserve (i) short transients (e.g., fault clicks) falling exactly on patch boundaries and (ii) loss of fine-grained intra-patch variance, which are particularly important in industrial acoustics.
>     - **Empirical Optimization (Stride 2 vs. Stride 1):** While stride-1 yields long token sequences with redundant spectral information, stride-2 reduces redundancy and computational cost while preserving transitional cues such as impulsive sound. Ablation confirms that stride-2 consistently improves performance across industrial tasks.
>
>     | Model | RenishawL (mAP) | VF2 (mAP) | Yornew (mAP) | ColdSpray (mAP) | Yornew (AUROC) | ColdSpray (AUROC) |
>     | :--- | :--- | :--- | :--- | :--- | :--- | :--- |
>     | IMPACT-DINOS (Stride-1) | 1.0000±0.0000  | 0.9669±0.0135  | 0.8660±0.0271  | 0.8740±0.0127  | 0.8832±0.0206  | 0.7442±0.0088  |
>     | IMPACT-DINOS (Stride-2) | 1.0000±0.0000  | 0.9746±0.0108  | 0.8841±0.0231  | 0.9125±0.0070  | 0.9121±0.0206  | 0.7852±0.0064  |
>
>     - **High-Resolution Upsampling Decoder:** While general audio relies on high-level semantics solvable with coarse patterns, industrial acoustics depend on fine-grained physical phenomena where subtle spectral textures indicate faults. IMPACT employs Transposed Convolution decoders to progressively upsample representations, enabling the reconstruction of detailed spectral structures (e.g., harmonics, sidebands, broadband noise). This design prioritizes the physical fidelity essential for diagnosis over the abstract semantic cues typically used in general audio.
>
> We will clarify these distinctions in Section 4 and de-emphasize any implication that we introduce a new SSL paradigm. Rather, our contribution is a domain-tailored adaptation supported by empirical evidence, combined with a dataset, benchmark, and physics-informed insights that together form a meaningful contribution for ICLR.

---

> ### Author Response · Authors · 2025-11-27
>
> **4. Empirical Comparison: IMPACT vs. EAT on DINOS**
>
> To further substantiate the architectural benefits discussed above, we conducted a direct performance comparison between IMPACT and the original EAT architecture. Both models were pretrained on the DINOS dataset under identical training configurations to ensure a fair comparison.
>
> While EAT shows a marginal advantage (+1.3%) in the relatively simpler VF2 task, IMPACT consistently outperforms EAT by a significant margin (ranging from 2.8% to 8.9%) in the more complex and physically demanding tasks (Yornew and ColdSpray). This empirical evidence confirms that our physics-aware architectural modifications—specifically the dual-stage encoding and high-resolution decoder—are highly effective for industrial acoustic perception.
>
> To ensure rigorous validation, we acknowledge that a comprehensive hyperparameter search for both architectures is necessary to rule out any optimization bias. We commit to conducting these extensive experiments and including the detailed analysis in the camera-ready version of the manuscript.
>
> | Model | RenishawL (mAP) | VF2 (mAP) | Yornew (mAP) | ColdSpray (mAP) | Yornew (AUROC) | ColdSpray (AUROC) |
> | :--- | :---: | :---: | :---: | :---: | :---: | :---: |
> | **IMPACT (Ours)** | **1.0000** | 0.9746±0.0108 | **0.8841±0.0231** | **0.9125±0.0070** | **0.9121±0.0206** | **0.7852±0.0064** |
> | EAT (Original) | **1.0000** | **0.9876±0.0055** | 0.8565±0.0257 | 0.8338±0.0076 | 0.8231±0.0240 | 0.7475±0.0087 |

---

> ### Author Response · Authors · 2025-11-27
>
> Dear Reviewer 66g5,
>
> As the discussion period is coming to a close, we would like to ensure that our response has adequately addressed your main concerns regarding evaluation fairness, model novelty, and the significance of our contribution.
>
> In our detailed rebuttal, we provided:
>
> - Evaluation Fairness: We conducted the requested experiment—pretraining AudioMAE from scratch on DINOS. The results confirm that IMPACT (\~18M) significantly outperforms AudioMAE (\~86M) even when both are pretrained on the same data (e.g., +11.9% mAP on ColdSpray).
>
> - Novelty & Superiority: We clarified the specific architectural advancements of IMPACT (Dual-stage CNN encoding, Stride-2 optimization, Upsampling reconstruction) compared to EAT. Furthermore, we provided a direct performance comparison (IMPACT vs. EAT) on DINOS, demonstrating that our domain-optimized architecture consistently outperforms the original EAT on complex industrial tasks.
>
> - Foundational Contribution: We elaborated on how this work establishes the first standardized ecosystem (Dataset, Reference Model, and Hierarchical Benchmark) for this field, addressing the critical lack of baselines that currently hinders reproducible research.
>
> We hope these additional experiments and clarifications resolve your concerns. We would be grateful if you could review our response and consider re-evaluating our submission.
>
> Thank you for your time and earnest review.
>
> Best regards,
>
> Authors

---

### Official Review · Reviewer_erTN · 2025-10-31

**Soundness:** 3
**Presentation:** 3
**Contribution:** 2
**Rating:** 6
**Confidence:** 4

**Summary:**

## An open-access dataset of industrial operation sounds

- Proposed DINOS, a dataset with over ~1000 hours of recordings from active manufacturing lines.
- Proposes IMPACT, a reference baseline models trained on the DINOS dataset.

**Strengths:**

- Paper is well written (except some minor grammatical errors. Authors, please recheck for missing spaces and punctuation.)
- A comprehensive benchmarking setup, with distinct pretraining and downstream benchmarking sets is provided.
- Limited availability of public, large-scale corpora is a major pain point in manufacturing and floor monitoring, so the dataset could indeed prove invaluable to the community.
- Evaluation, to the extent done in the paper, is good.

**Weaknesses:**

- Based on the results alone, it is hard to say how useful the proposed dataset is over the publicly available DCASE2025 Challenge Task 2 dataset for pretraining.

**Questions:**

1. Is there an overlap between the pretraining set for DINOS and DCASE2025 Challenge Task 2?
2. Why is your paper titled after the model, and not the dataset?

---

> ### Author Response · Authors · 2025-11-22
>
> We sincerely thank you for your constructive feedback and for recognizing the potential value of our dataset to the community. We will carefully proofread the camera-ready version to fix the minor grammatical and formatting issues you pointed out. Below, we address your concerns regarding the utility of DINOS compared to DCASE, data overlap, and the paper title.
>
> ---
>
> ### **1. Utility of DINOS over DCASE2025 Challenge Task 2 Dataset**
>
> While DCASE Task 2 has been valuable for studying generic machine-condition monitoring, its focus on small, controlled setups makes it only weakly related to the complex acoustic environments encountered on real shop floors. DINOS is designed to fill exactly this gap. The performance gain stems from DINOS capturing real physics-driven acoustic signatures, such as tool-workpiece interaction under production forces and arbitrary background noise patterns, which are difficult to replicate in controlled settings. This broader physical variability enables generalization to unseen machines and domains, extending beyond simple anomaly detection.
>
> **Fine-Grained Classification with Limited Training Data (Yornew Classification, T6-T17)**
>
> Discriminating among 12 distinct machining states varying by cutting depth, speed, and material removal rate.
>
> - IMPACT-DINOS: 0.8841 mAP (+5.3%)
> - IMPACT-DCASE: 0.8316 mAP
> - The high-fidelity industrial recordings in DINOS allow the model to learn sharper decision boundaries for fine-grained operational states compared to the general DCASE dataset.
>
> **Anomaly Detection with Limited Training Data (Yornew Anomaly Detection, T26)**
>
> Detecting chatter (a subtle self-excited vibration anomaly) which requires high sensitivity to fine-grained spectral textures.
>
> - IMPACT-DINOS: 0.9121 AUROC (+13.4%)
> - IMPACT-DCASE: 0.7777 AUROC
> - Pretraining on an industrial dataset enhances diagnostic sensitivity.
>
> **Classification in Unseen Domain (ColdSpray Classification, T18-T25)**
>
> Operational status classification in a domain completely unseen during pretraining, involving supersonic fluid phenomena (gas flow, powder impact) rather than typical mechanical operation.
>
> - IMPACT-DINOS: 0.9125 mAP (+7.8%)
> - IMPACT-DCASE: 0.8344 mAP
> - DINOS enables the model to learn robust physical acoustic features that transfer effectively to novel machinery and physics.
>
> **Anomaly Detection in Unseen Domain (ColdSpray Anomaly Detection, T27)**
>
> Detecting anomalies in a domain completely unseen during pretraining, involving supersonic fluid phenomena (gas flow, powder impact) rather than typical mechanical operation.
>
> - IMPACT-DINOS: 0.7852 AUROC (+6.8%)
> - IMPACT-DCASE: 0.7176 AUROC
> - DINOS improves the model's capability to extract meaningful physical signals even under an unseen domain.
>
> By incorporating industrial acoustics, DINOS allows the model to better grasp the intrinsic acoustic characteristics of industrial equipment, thereby demonstrating superior generalization and performance even on machines and domains not encountered during training.
>
> ---
>
> ### **2. Clarification on Data Overlap**
>
> No, there is absolutely no overlap. We maintained a strict separation between the datasets. The DINOS pretraining set consists exclusively of data we collected from live shop floors (7,404 recordings, 121 hours), which is entirely distinct from the DCASE2025 corpus. Furthermore, even within our own data, we strictly separated the pretraining set (e.g., RenishawR) from the benchmarking set (e.g., RenishawL) to ensure rigorous evaluation.
>
> ---
>
> ### **3. Paper Title**
>
> We titled the paper IMPACT rather than DINOS because our contribution is an integrated benchmark framework, not merely a dataset release.
>
> The three components are interdependent:
>
> 1. DINOS: 1,093 hours of shop floor recordings
> 2. IMPACT: Standardized reference model
> 3. Evaluation protocol: 27-task progressive benchmark
>
> DINOS alone would be unprocessed audio files without a clear evaluation methodology. IMPACT provides the missing standardization layer that enables reproducible comparison, similar to how ImageNet's impact came from the combination of data and standardized evaluation, not just the images themselves. However, we fully agree with your assessment that DINOS represents the most novel component, as it addresses the community's need for real industrial acoustic data. We will revise the abstract and introduction to establish DINOS prominently in the opening sentences, clarifying that IMPACT serves as the reference framework to unlock DINOS's value.
>
> ---
>
> We believe this combination of real-world data (DINOS) with standardized evaluation (IMPACT) addresses the critical gap between laboratory research and industrial deployment, as evidenced by the consistent performance improvements across evaluation scenarios.

---

> ### Comment · Reviewer_erTN · 2025-11-26
> **Response to the authors**
>
> Thanks for the clarification. I had read the author's rebuttal on the very day it was posted, and I did not have any further questions or actionable items.
>
> However, seeing the discussion and the following specific comments made by the other reviewers:
>
> By Reviewer US9t
>
> > The paper has limited novelty. The primary contribution of the paper is the dataset; the IMPACT model is based on a well-known existing self-supervised model, EAT.
>
> and the following comments by Reviewer 66g5
>
> > The evaluation is critically insufficient and cannot show the superiority of IMPACT. The authors did not apply other pretraining methods (e.g., AudioMAE) on DINOS. They only evaluated the off-the-shelf pretrained models (e.g., a model pretrained using AudioMAE method on other acoustic datasets) on DINOS. Since IMPACT is a pretraining method, if the authors want to show the superiority of IMPACT, they need to pretrain IMPACT and other pretraining methods (e.g., AudioMAE) on the same datasets.
>
> > Therefore, if the majority of the contributions lie in the introduction of DINOS, then this paper might be below the bar of ICLR. *It might be more suitable to submit this paper to a venue specialized in industrial sensing or a venue offering dataset tracks*
>
> I would like to state the following:
>
> **The paper is a datasets and benchmark paper**, and in my opinion, has more than sufficient contribution and it indeed addresses a critical gap in the realm of machine-condition monitoring. In my opinion, we need to support the addition of more datasets in the domain. I think the other reviewers' comments regarding the lack of contribution and novelty are unnecessarily harsh.
>
> I strongly support the inclusion of this paper within datasets and benchmarks: I believe it has the potential to be a very valuable contribution to the community in the future, and while ICLR does not have a dedicated Datasets track akin to NeurIPS, I think it still warrants inclusion. To show this support, I will increase my score.

---

> > ### Author Response · Authors · 2025-11-26
> >
> > We sincerely appreciate your recognition of our core motivation to address the critical gap in machine-condition monitoring caused by the scarcity of large-scale public datasets and the absence of standardized benchmarking protocols.
> >
> > As you rightly noted, establishing a solid foundation is essential to fostering community-driven progress in this domain. To fully support this goal, we commit to including publicly accessible links in the camera-ready version for:
> > 1. The Complete DINOS dataset with comprehensive documentation.
> > 2. The IMPACT source code and pretrained checkpoints.
> > 3. The Benchmark evaluation scripts and preprocessing pipelines.
> >
> > Thank you again for your strong support of datasets and benchmarks in advancing this field.

---

### Official Review · Reviewer_US9t · 2025-11-01

**Soundness:** 2
**Presentation:** 2
**Contribution:** 2
**Rating:** 4
**Confidence:** 2

**Summary:**

The paper proposes DINOS (Diverse INdustrial Operation Sounds), a large-scale dataset for understanding Industrial acoustic signals at a large scale. The paper also trained a self-supervised baseline model on the data IMPACT (Industrial Machine Perception via Acoustic Cognitive Transformer).

**Strengths:**

The paper is unique and interesting. The paper is written well and contains detailed experiments. The self-supervised model IMPACT, trained on the proposed data, achieves the best performance across the majority of the tasks.

**Weaknesses:**

The paper has limited novelty. The primary contribution of the paper is the dataset; the IMPACT model is based on a well-known existing self-supervised model, EAT.

**Questions:**

What is the number of parameters across various models in Table 4? Does the Impact model work better because it is a larger model, or due to pretraining on DINOS?

---

> ### Author Response · Authors · 2025-11-22
>
> We thank you for finding our work unique and interesting. We also appreciate your feedback regarding the model's novelty and architecture. Below, we clarify the specific architectural advancements of IMPACT compared to EAT.
>
> ---
>
> **1. Model Size and Performance Analysis**
>
> IMPACT is not larger than the other pretrained baselines.
>
> - Parameter Count:
>     - IMPACT: ~18M
>     - AudioMAE: ~86M
>     - VGGish: ~62M
>
> IMPACT outperforms baselines despite being 1/5th the size. These improvements cannot be attributed to model size, but rather to (i) pretraining on domain-specific industrial data (DINOS) and (ii) the physics-aware architectural optimizations described below.
>
> ---
>
> **2. Architectural Distinctiveness (IMPACT vs. EAT)**
>
> While IMPACT shares the efficient distillation philosophy of EAT, it employs a distinct hybrid architecture engineered specifically to capture the physical characteristics of industrial machine sounds. In particular, the CNN components in IMPACT play different roles than in EAT:
>
> - **Dual-Stage Encoding (vs. EAT's Single Patching)**
>     - **Physics-Aware Feature Extraction**: EAT uses a CNN encoder mainly as a non-overlapping patch embedder (kernel size equals stride), so each token is computed from a disjoint spectro-temporal region. In IMPACT, we introduce an additional CNN feature extractor with overlapping receptive fields before patch embedding. This design is intended to reduce issues such as (i) short transients (e.g., fault clicks) falling exactly on patch boundaries and (ii) loss of fine-grained intra-patch variance, which are particularly important in industrial acoustics.
>     - **Empirical Optimization of Stride 2 vs. Stride 1**: Our preliminary experiments indicated that a stride-2 configuration yields a better performance compared to stride 1. While stride 1 preserves full spatial resolution, it produces long token sequences with redundant spectral information, making the model more sensitive to small sensor-level fluctuations and increasing the quadratic cost of self-attention. Using stride 2 acts as a mild noise filtering and downsampling step. It reduces redundancy and sequence length, alleviates the computational cost of self-attention, and enables the model to focus on meaningful changes.
>
> | Model | RenishawL (mAP) | VF2 (mAP) | Yornew (mAP) | ColdSpray (mAP) | Yornew (AUROC) | ColdSpray (AUROC) |
> | :--- | :--- | :--- | :--- | :--- | :--- | :--- |
> | IMPACT-DINOS (Stride-1) | 1.0000±0.0000  | 0.9669±0.0135  | 0.8660±0.0271  | 0.8740±0.0127  | 0.8832±0.0206  | 0.7442±0.0088  |
> | IMPACT-DINOS (Stride-2) | 1.0000±0.0000  | 0.9746±0.0108  | 0.8841±0.0231  | 0.9125±0.0070  | 0.9121±0.0206  | 0.7852±0.0064  |
>
> - **High-Resolution Upsampling Decoder:**
>     General audio is typically characterized by high-level semantics. In these settings, downstream tasks can often be solved from relatively coarse-grained time–frequency patterns. In contrast, industrial acoustic labels are tightly coupled to fine-grained physical phenomena, such as short impulsive events within harmonic lines. As a result, small variations in spectral texture can directly correspond to changes in machine condition or the presence of an incipient fault. To effectively capture these textures, IMPACT employs Transposed Convolution decoders that progressively upsample the latent representation back to a high-resolution spectrogram. This design reconstructs detailed spectral structure, including harmonic patterns, sidebands, and broadband machine noise. This is important for industrial acoustics, where such structures are often more informative about the underlying physical state than high-level semantic cues. In our experiments, this decoder achieved better performance on the benchmark tasks, and we will clarify this motivation and design in the revised manuscript.

---

> ### Author Response · Authors · 2025-11-22
>
> **3. Broader Contribution and Novelty Beyond the Dataset - Establishing the AI-driven Industrial Acoustic Perception Ecosystem**
>
> We agree that DINOS is the most visible component of our work. At the same time, our goal is not only to release a dataset, but to establish a comprehensive foundation for AI-driven industrial acoustic perception that the community can build upon. In particular, our contributions extend beyond the raw data in three ways:
>
> - **Standardized Hierarchical Benchmark Methodology**: We propose a 4-step, 6-type hierarchical benchmark that ranges from simple on/off discrimination to cross-machine and cross-sensor domain shifts. This is, to our knowledge, the first attempt to organize industrial acoustic evaluation along increasing levels of difficulty and domain mismatch, rather than as a single ad-hoc task. We believe this provides a clear rationale for testing future models under realistic deployment scenarios, not just on matched-condition anomaly detection.
> - **Physics-Informed Dataset Design Insights**: Our experiments indicate that coverage over underlying physical phenomena (e.g., frictional contact, impact, fluid flow) is more important for generalization than simply increasing the number of machines. For example, the surprisingly strong performance of OPERA (pretrained on respiratory sounds) on ColdSpray tasks suggests that pretraining on acoustics dominated by similar fluid-dynamic patterns can sometimes be more beneficial than pretraining on mechanically different machines. We present these observations as design guidelines for future industrial audio datasets in the conclusion section.
> - **A Complete Reference Foundation**: By integrating DINOS (data), IMPACT (reference model), and the hierarchical benchmark protocol, we provide a complete, reusable testbed. Researchers can plug in new pretraining methods and architectures and directly compare them under the same physically grounded tasks and evaluation procedure. In this sense, the novelty of our work lies in establishing the first standardized, domain-specific foundation for industrial acoustic representation learning, rather than in claiming a completely new SSL paradigm.
>
> ---
>
> To further support the community and ensure reproducibility, we will publicly release the DINOS dataset, the IMPACT pretrained checkpoints, and all benchmark code and evaluation scripts upon publication.

---

> ### Author Response · Authors · 2025-11-27
>
> **4. Empirical Comparison: IMPACT vs. EAT on DINOS**
>
> To provide concrete evidence of the architectural benefits discussed above, we conducted a direct performance comparison between IMPACT and the original EAT architecture. Both models were pretrained on the DINOS dataset under identical training configurations to ensure fairness.
>
> The results demonstrate the distinct advantage of IMPACT's domain-optimized architecture. While EAT shows a slight marginal advantage (+1.3%) in the relatively simpler VF2 task, IMPACT consistently outperforms EAT by a significant margin ranging from 2.8% to 8.9% in the more complex and physically demanding tasks (Yornew and ColdSpray). This empirical evidence confirms that our architectural modifications are highly effective in capturing the fine-grained physical textures of industrial sounds.
>
> To ensure the most rigorous comparison, we acknowledge that a comprehensive grid search for both architectures is necessary. We will conduct these extensive experiments and including the detailed analysis in the camera-ready version of the manuscript.
>
> | Model | RenishawL (mAP) | VF2 (mAP) | Yornew (mAP) | ColdSpray (mAP) | Yornew (AUROC) | ColdSpray (AUROC) |
> | :--- | :---: | :---: | :---: | :---: | :---: | :---: |
> | **IMPACT (Ours)** | **1.0000** | 0.9746±0.0108 | **0.8841±0.0231** | **0.9125±0.0070** | **0.9121±0.0206** | **0.7852±0.0064** |
> | EAT (Original) | **1.0000** | **0.9876±0.0055** | 0.8565±0.0257 | 0.8338±0.0076 | 0.8231±0.0240 | 0.7475±0.0087 |

---

> ### Author Response · Authors · 2025-11-27
>
> Dear Reviewer US9t,
>
> As the discussion period is coming to a close, we would like to confirm if our response has clarified your questions regarding model size, novelty, and the scope of our contribution.
>
> In our rebuttal, we provided:
>
> - Parameter Comparison: Clarifying that IMPACT (\~18M) is 1/5th the size of the baselines (\~86M), proving that performance gains stem from architectural efficiency, not model capacity.
>
> - Architectural Distinctiveness: Detailing the physics-aware modifications (Dual-stage CNN encoding, Stride-2 optimization, Upsampling reconstruction) that distinguish IMPACT from EAT, supported by new empirical comparisons (IMPACT vs. EAT).
>
> - Foundational Contribution: We highlighted that our contribution extends beyond the model itself to establishing a comprehensive research ecosystem—including a standardized hierarchical benchmark and physics-informed design guidelines—which serves as a necessary foundation for future advancements in industrial acoustic perception.
>
> We hope these clarifications and additional results resolve your concerns. We would be grateful if you could review our response and consider updating your assessment accordingly.
>
> Thank you for your valuable feedback.
>
> Best regards,
>
> Authors

---

### Author Response · Authors · 2025-11-29
**Final Remarks**

Dear Area Chair,

We sincerely thank the reviewers for their constructive feedback. We submit this final summary to highlight that **(1) all reviewers acknowledge the foundational value of this work**, **(2) all major concerns regarding evaluation fairness and novelty have been definitively resolved through new experiments**, and **(3) this work provides critical infrastructure for an underserved research domain**.

---

## Summary for Area Chair

| Aspect | Status |
| --- | --- |
| **Reviewer Consensus** | All reviewers recognize the dataset's value; Reviewer erTN explicitly raised their score in support |
| **Evaluation Fairness (66g5)** | Fully addressed with AudioMAE-Scratch experiment on DINOS |
| **Model Novelty (US9t, 66g5)** | Clarified with IMPACT vs. EAT comparison and architectural distinctions |
| **Contribution Scope** | Integrated ecosystem: Dataset + Reference Model + Hierarchical Benchmark |
| **Community Impact** | First large-scale, real-world industrial acoustic foundation for reproducible research |

---

## 1. Reviewer Consensus on Foundational Value

All three reviewers acknowledge the significance of addressing the critical gap in industrial acoustic perception. Reviewer erTN, after considering other reviewers' comments, explicitly stated:

> *"The paper is a datasets and benchmark paper... indeed addresses a critical gap... I think the other reviewers' comments regarding the lack of contribution and novelty are unnecessarily harsh. I strongly support the inclusion of this paper within datasets and benchmarks... to be a very valuable contribution to the community... To show this support, I will increase my score."*

Our contribution extends beyond raw data release to establish a complete research ecosystem:

- **DINOS:** 1,093 hours of the first large-scale industrial acoustic dataset collected from real production environments
- **IMPACT:** A pretrained reference model enabling standardized comparison
- **Hierarchical Benchmark Protocol:** A 4-step, 27-task evaluation framework progressing from simple on/off detection to cross-machine and cross-sensor domain shifts

This integrated approach addresses the long-standing absence of standardized foundations that has hindered reproducible, community-driven research in industrial acoustic perception.

---

## 2. Utility of DINOS over DCASE2025 (Reviewer erTN)

We demonstrated that DINOS captures real physical signatures lacking in DCASE, enabling IMPACT-DINOS to outperform IMPACT-DCASE across all challenging benchmarks:

| Task | Metric | IMPACT-DCASE | IMPACT-DINOS | Δ |
| --- | --- | --- | --- | --- |
| Fine-Grained Classification (Yornew) | mAP | 0.8316 | **0.8841** | +5.3% |
| Anomaly Detection (Yornew) | AUROC | 0.7777 | **0.9121** | +13.4% |
| Unseen Domain Classification (ColdSpray) | mAP | 0.8344 | **0.9125** | +7.8% |
| Unseen Domain Anomaly Detection (ColdSpray) | AUROC | 0.7176 | **0.7852** | +6.8% |

These results confirm that DINOS provides domain-specific acoustic characteristics essential for industrial perception tasks that cannot be obtained from existing public corpora.

---

## 3. Model Size and Efficiency (Reviewer US9t)

We clarified that IMPACT's performance gains stem from architectural efficiency and domain-specific pretraining, not model capacity:

| Model | Parameters |
| --- | --- |
| IMPACT | ~18M |
| AudioMAE | ~86M |
| VGGish | ~62M |

IMPACT achieves superior performance despite being **1/5th the size** of AudioMAE, demonstrating that our physics-aware architectural modifications are highly effective for industrial acoustics.

---

## 4. Evaluation Fairness: AudioMAE Trained from Scratch (Reviewer 66g5)

We conducted the exact experiment requested: pretraining AudioMAE from scratch on DINOS under identical conditions. This ensures a fair comparison where both models have equal access to domain-specific data.

| Model | RenishawL (mAP) | VF2 (mAP) | Yornew (mAP) | ColdSpray (mAP) | Yornew (AUROC) | ColdSpray (AUROC) |
| --- | --- | --- | --- | --- | --- | --- |
| AudioMAE-PreT. (AudioSet) | 0.9959±0.0024 | 0.9418±0.0166 | 0.5743±0.0167 | 0.7370±0.0141 | 0.6224±0.0148 | 0.4371±0.0129 |
| AudioMAE-FineT. (DINOS) | 0.9866±0.0194 | 0.9655±0.0196 | 0.6890±0.0338 | 0.8370±0.0100 | 0.7000±0.0363 | 0.6316±0.0186 |
| AudioMAE-Scratch (DINOS) | 1.0000±0.0000 | 0.9482±0.0186 | 0.7729±0.0412 | 0.7932±0.0136 | 0.7238±0.0187 | 0.5221±0.0061 |
| **IMPACT-DINOS** | **1.0000±0.0000** | **0.9746±0.0108** | **0.8841±0.0231** | **0.9125±0.0070** | **0.9121±0.0206** | **0.7852±0.0064** |

- **DINOS is effective:** Both AudioMAE-Scratch and AudioMAE-FineT. outperform AudioMAE-PreT., confirming DINOS provides valuable industrial-specific acoustic characteristics.
- **IMPACT's architecture matters:** Even under identical pretraining conditions, IMPACT significantly outperforms AudioMAE, proving that performance gains are attributable to our domain-optimized architecture, not merely data access.

---

---

> ### Author Response · Authors · 2025-11-29
> **Final Remarks**
>
> ## 5. Architectural Novelty: IMPACT vs. EAT (Reviewers US9t, 66g5)
>
> While IMPACT builds upon EAT's student–teacher framework, it introduces physics-aware modifications specifically designed for industrial acoustics:
>
> | Component | EAT (Original) | IMPACT (Ours) | Rationale |
> | --- | --- | --- | --- |
> | **Encoding** | Single non-overlapping patching (kernel = stride) | Dual-stage CNN with overlapping receptive fields | Preserves short transients on patch boundaries; captures fine-grained intra-patch variance critical for fault detection |
> | **Stride** | Single non-overlapping patching (kernel = stride) | Stride-2 optimization | Reduces redundancy and computational cost while preserving transitional cues |
> | **Decoder** | Standard CNN decoder | Upsampling Transposed Conv decoder | Reconstructs detailed spectral structures essential for physical state diagnosis |
>
> To provide concrete evidence, we conducted a direct comparison between IMPACT and EAT, both pretrained on DINOS:
>
> | Model | RenishawL (mAP) | VF2 (mAP) | Yornew (mAP) | ColdSpray (mAP) | Yornew (AUROC) | ColdSpray (AUROC) |
> | --- | --- | --- | --- | --- | --- | --- |
> | EAT (Original) | **1.0000±0.0000** | **0.9876±0.0055** | 0.8565±0.0257 | 0.8338±0.0076 | 0.8231±0.0240 | 0.7475±0.0087 |
> | **IMPACT (Ours)** | **1.0000±0.0000** | 0.9746±0.0108 | **0.8841±0.0231** | **0.9125±0.0070** | **0.9121±0.0206** | **0.7852±0.0064** |
>
> While EAT shows marginal advantage on the simpler VF2 task, **IMPACT consistently outperforms EAT by 2.8%–8.9%** on complex, physically demanding tasks (Yornew and ColdSpray), confirming our physics-aware modifications are highly effective.
>
> Additionally, our stride ablation validates the design choice:
>
> | Model | Yornew (mAP) | ColdSpray (mAP) | Yornew (AUROC) | ColdSpray (AUROC) |
> | --- | --- | --- | --- | --- |
> | IMPACT (Stride-1) | 0.8660±0.0271 | 0.8740±0.0127 | 0.8832±0.0206 | 0.7442±0.0088 |
> | **IMPACT (Stride-2)** | **0.8841±0.0231** | **0.9125±0.0070** | **0.9121±0.0206** | **0.7852±0.0064** |
>
> ---
>
> ## 6. Novel Finding: Physics-Informed Design Guidelines
>
> Our benchmarking revealed an important insight: **generalization depends more on underlying physical phenomena than on machine types**. The surprising success of OPERA (trained on respiratory sounds) on ColdSpray tasks, which involve fluid dynamics similar to respiratory events, highlights that pretraining datasets should be curated based on physical sound-generation mechanisms, not merely machine diversity.
>
> We distill these findings into design guidelines for future industrial datasets, proposing a paradigm shift from simple data collection to physics-aware curation. This represents a novel contribution to the methodology of industrial acoustic research.
>
> ---
>
> ## 7. Venue Appropriateness
>
> While ICLR does not have a dedicated track, the conference has the Datasets and Benchmark Area. Moreover, ICLR has historically embraced foundational contributions that enable representation learning research. Our work establishes the necessary infrastructure for advancing self-supervised learning in industrial acoustics, which is an underserved domain with significant real-world impact. The integrated nature of our contribution (Dataset + Model + Benchmark) aligns with ICLR's mission to advance representation learning methodologies.
>
> ---
>
> ## Conclusion
>
> We respectfully submit that this work makes a significant and timely contribution:
>
> | Aspect | Contribution |
> | --- | --- |
> | **Dataset Novelty** | First large-scale, real-world industrial acoustic dataset (DINOS: 1,093 hours) |
> | **Methodological Novelty** | Standardized hierarchical benchmark with 27 tasks across 4 difficulty levels |
> | **Technical Contribution** | Physics-aware reference model (IMPACT) with demonstrated superiority over both general-purpose models and original EAT architecture |
> | **Empirical Rigor** | Comprehensive ablations, fair comparisons (AudioMAE trained from scratch), and direct IMPACT vs. EAT evaluation—all in response to reviewer requests |
> | **Community Value** | Public release of dataset, pretrained checkpoints, and benchmark code |
>
> **Given that (1) all reviewers acknowledge the foundational value of this work, (2) we have fully addressed the primary concerns regarding evaluation fairness and novelty with new experiments, and (3) Reviewer erTN explicitly increased their score in support, we respectfully request the Area Chair to consider acceptance of this submission.**
>
> We are committed to releasing all resources upon publication to support reproducible research in this underserved domain.
>
> Best regards,
>
> The Authors

---

### Meta-Review · Area_Chair_KAiF · 2026-01-07

**Summary:**

I share the reviewers’ concerns that the paper has limited conceptual and technical novelty. However, I also agree with reviewer erTN in that the paper largely accomplishes what it sets out to as a datasets and benchmarks contribution. So, I do not consider this alone reason enough to reject it. Leaving the issue of novelty alone, I believe this is a fairly well-designed data collection effort and benchmark design.

By far the biggest issue, which I don’t think has been adequately discussed, is that the benchmark seems quite close to saturation. It’s not clear how the community is supposed to get much mileage out of the benchmark given these conditions (i.e. it seems too easy and various strong audio representations score highly). Even VGGish, which is an incredibly flawed representation, scores highly on a couple of them currently. Only T7, T8 and T27, interpreting generously, seem to have enough headroom for there to be a problem to solve by future models, from the results. Without a challenging benchmark to build on, the contribution falls back to the IMPACT modeling contribution, which does not seem sufficient on its own to achieve the projected impact on this community.

I would also make a few additional suggestions:
1. The “Cognitive” in the title is puzzling. It’s very unclear what is “cognitive” about any of this. If the goal is to backronym from IMPACT, I’m sure a better C word can be found.
2. Tables 3/4 are *way too* small. Please think aboutsplitting them into multiple tables for clarity, or re-organizing

**Reviewer Concerns:**

### Meaningfully addressed by the author response:
- **US9t**: W1 limited novelty (partially addressed, well-addressed by reviewer erTN); Q1 parameters
- **erTN**: Q1 overlap with DCASE Task 2 (addressed)
- **66g5**: W1 AudioMAE (addressed with extra experiment), limited novelty (addressed in principle, although I believe the novelty of these advances is low and domain constrained)

### Remaining concerns:
- **erTN**: W1 utility over DCASE Task 2 (the improvements here could simply be related to distribution shift, and do not demonstrate greater utility. What are the results on DCASE Task 2? Does DINOS also help with that?); Q2 (paper title — IMPACT is both the model and the framework? Something is unclear here, but it’s a very minor concern IMO)

**Reviewer Scores:**

This is difficult to predict except for reviewer erTN who explicitly indicated intent to raise their score. I don’t think the other reviewers have been given enough cause for a meaningful score increase.

---

### Decision · Program_Chairs · 2026-01-26

Reject